# Kainate receptor channel opening and gating mechanism

Shanti Pal Gangwar[1,5], Maria V. Yelshanskaya[1,5], Kirill D. Nadezhdin[1,5], Laura Y. Yen[1,2], Thomas P. Newton[1,3], Muhammed Aktolun[4], Maria G. Kurnikova[4] & Alexander I. Sobolevsky[1✉]

Kainate receptors, a subclass of ionotropic glutamate receptors, are tetrameric ligand-gated ion channels that mediate excitatory neurotransmission[1–4]. Kainate receptors modulate neuronal circuits and synaptic plasticity during the development and function of the central nervous system and are implicated in various neurological and psychiatric diseases, including epilepsy, depression, schizophrenia, anxiety and autism[5–11]. Although structures of kainate receptor domains and subunit assemblies are available[12–18], the mechanism of kainate receptor gating remains poorly understood. Here we present cryo-electron microscopy structures of the kainate receptor GluK2 in the presence of the agonist glutamate and the positive allosteric modulators lectin concanavalin A and BPAM344. Concanavalin A and BPAM344 inhibit kainate receptor desensitization and prolong activation by acting as a spacer between the amino-terminal and ligand-binding domains and a stabilizer of the ligand-binding domain dimer interface, respectively. Channel opening involves the kinking of all four pore-forming M3 helices. Our structures reveal the molecular basis of kainate receptor gating, which could guide the development of drugs for treatment of neurological disorders.

Kainate receptors (KARs) are members of the ionotropic glutamate receptor family of tetrameric ion channels[1,2]. Together with AMPA (α-amino-3-hydroxy-5-methyl-4-isoxazole propionic acid) receptors (AMPARs) and N-methyl-D-aspartate (NMDA) receptors, postsynaptic KARs mediate excitatory neurotransmission in the central nervous system[3,4]. In addition, presynaptic KARs act as neuromodulators by tuning the secretion of excitatory and inhibitory neurotransmitters[4,19]. KARs regulate the development and maturation of neuronal connections and are involved in memory formation. They are implicated in neurodevelopmental, neurodegenerative and psychiatric disorders, including pain, epilepsy, migraine, Huntington's disease, schizophrenia, Down syndrome, depression, and bipolar, autism spectrum and obsessive-compulsive disorders, therefore representing an important target for therapeutic development[5–11].

KAR tetramers are assembled from a pool of five subunits, GluK1–5. GluK1–3 can form functional homotetrameric channels, whereas GluK4 and GluK5 subunits must co-assemble with GluK1–3 to form functional heterotetramers[20]. KAR tetramers can also bind NETO1 and NETO2, auxiliary subunits that influence their synaptic trafficking[21–23]. Both principal and auxiliary subunits determine KAR functional properties, including ion permeation, channel block, ligand specificity and kinetics, and have important roles in shaping KAR-mediated synaptic responses[24–32]. Each KAR subunit has a multi-domain organization, including the amino-terminal domain (ATD), involved in receptor assembly and trafficking, the ligand-binding domain (LBD) that binds agonists, competitive antagonists and positive allosteric modulators (PAMs), the transmembrane domain (TMD), which forms an ion-conducting channel, and the carboxy-terminal domain, which determines surface expression, synaptic localization and interactions with intracellular proteins.

Isolated KAR ATD and LBD structures revealed their clamshell architecture and mechanisms of ligand binding[33,34], exhibiting high similarity to AMPARs[35–37]. Structures of full-length GluK2 and GluK3 homotetramers and GluK2–GluK5 heterotetramers have been resolved in the apo, agonist-bound and competitive-antagonist-bound states[12–18]. Although many of these structures showed KAR domain assembly analogous to AMPARs, structures in the apo and agonist-bound states demonstrated a nearly four-fold symmetrical arrangement of LBDs, which is non-canonical for ionotropic glutamate receptors. Given the unique role of KARs in the central nervous system, their gating is also expected to differ from that of other ionotropic glutamate receptors. Although AMPAR gating has been revealed by open-state structures in the presence of agonist, PAM and activating auxiliary subunit γ2 (refs. 38,39) (also known as Stargazin), open-state structures of KARs have remained unresolved.

Here we performed structural studies of KAR gating using time-resolved cryo-electron microscopy (cryo-EM). Fast application of agonist glutamate (Glu) in the presence of the PAMs lectin concanavalin A (ConA) and BPAM344 (hereafter BPAM) enabled us to trap an ensemble of GluK2 KAR conformations, including the fully-liganded open state and Glu-bound non-conducting states. These structures reveal

[1]Department of Biochemistry and Molecular Biophysics, Columbia University, New York, NY, USA. [2]Cellular and Molecular Physiology and Biophysics Graduate Program, Columbia University Irving Medical Center, New York, NY, USA. [3]Integrated Program in Cellular, Molecular and Biomedical Studies, Columbia University Irving Medical Center, New York, NY, USA. [4]Department of Chemistry, Carnegie Mellon University, Pittsburgh, PA, USA. [5]These authors contributed equally: Shanti Pal Gangwar, Maria V. Yelshanskaya, Kirill D. Nadezhdin. ✉e-mail: as4005@cumc.columbia.edu

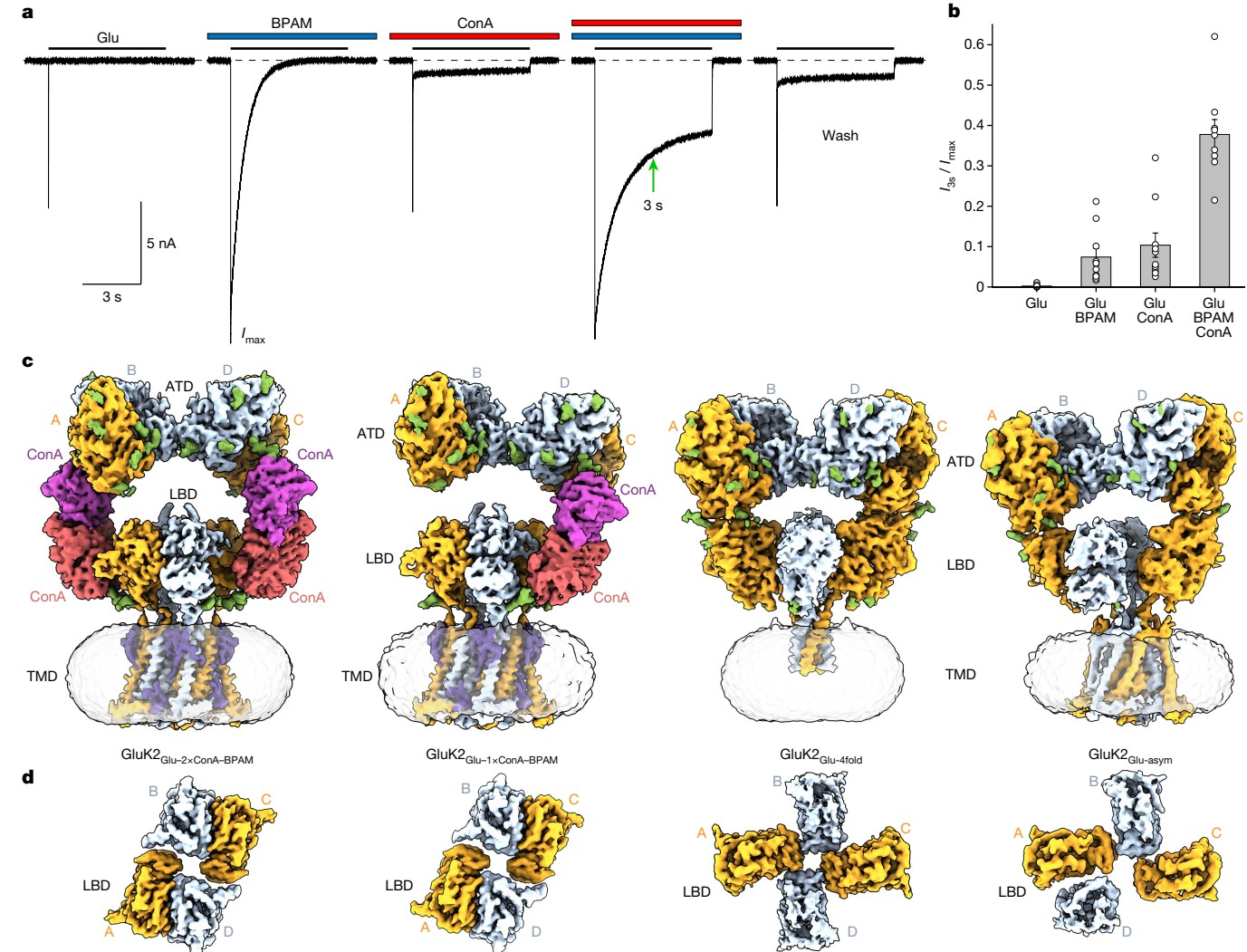

**Fig. 1 | KAR function and time-resolved cryo-EM. a**, Whole-cell patch-clamp currents recorded at −60 mV membrane potential from HEK 293F cells expressing rat GluK2 in response to 6-s application of 3 mM Glu alone, in the continuous presence of 50 µM BPAM or 0.3 mg ml⁻¹ ConA, or in the presence of both BPAM and ConA. The green arrow indicates the 3-s mark since the beginning of Glu application. **b**, GluK2-mediated current measured at 3 s after the beginning of Glu application ($I_{3s}$) as a fraction of the maximum current ($I_{max}$).

Data are mean ± s.e.m. Biologically independent measurements: $n = 10$ for Glu, $n = 10$ for Glu + BPAM, $n = 10$ for Glu + ConA and $n = 9$ for Glu + BPAM + ConA. Data are representative of three independent experiments. **c**, Time-resolved 3D cryo-EM reconstructions of GluK2 captured within 3 s of Glu application in the continuous presence of ConA and BPAM. **d**, Extracellular view of density for the LBD layer in the same reconstructions as in **c**.

the unique gating mechanism of KARs, uncover the mechanism of their regulation by ConA[40–42], provide molecular templates for understanding their distinct roles in the central nervous system, and suggest potential strategies for drug development to treat numerous neurological and psychiatric diseases.

## Synergistic effect of PAMs on GluK2 function

We tested KAR function using patch-clamp current recordings from HEK 293F cells expressing rat GluK2. At a membrane potential of −60 mV, prolonged application of 3 mM Glu elicited a rapidly activating inward current that further decayed with a time constant $\tau$ of $3.89 \pm 0.27$ ms ($n = 10$) to nearly zero value ($I_{3s}/I_{max} = 0.0022 \pm 0.0011$, $n = 10$) owing to receptor desensitization (Fig. 1a, first trace). In the presence of 50 µM BPAM, desensitization was 270 times slower (desensitization time constant ($\tau$) = $1.05 \pm 0.13$ s, $n = 10$; Fig. 1a, second trace) but a 3 s exposure to Glu also resulted in substantial current reduction ($I_{3s}/I_{max} = 0.074 \pm 0.021$, $n = 10$). In the presence

of 0.3 mg ml⁻¹ ConA, fast desensitization ($\tau = 9.08 \pm 1.15$ ms, $n = 10$) was accompanied by an even slower component ($\tau = 2.64 \pm 0.23$ s, $n = 10$) (Fig. 1a, third trace). However, the relatively small fraction of the slow component ($9.1 \pm 3.2\%$, $n = 10$) did not prevent considerable current reduction on the 3-s time scale ($I_{3s}/I_{max} = 0.104 \pm 0.03$, $n = 10$).

In the presence of both ConA and BPAM, desensitization included only the slow component ($\tau = 1.94 \pm 0.17$ s, $n = 10$; Fig. 1a, fourth trace). Within 3 s of Glu application, the current amplitude was reduced to less than one-third of its maximal value ($I_{3s}/I_{max} = 0.378 \pm 0.037$, $n = 10$). After ConA and BPAM washout, the Glu-induced current (Fig. 1a, fifth trace) resembled the current recorded in the continuous presence of ConA alone (Fig. 1a, third trace), suggesting that BPAM dissociates from GluK2 readily, whereas ConA remains bound. Given the synergistic effect of PAMs on the current amplitude (Fig. 1b), we reasoned that more than one-third of GluK2 receptors can be trapped in the open state by an exposure to Glu for less than 3 s in the continuous presence of ConA and BPAM.

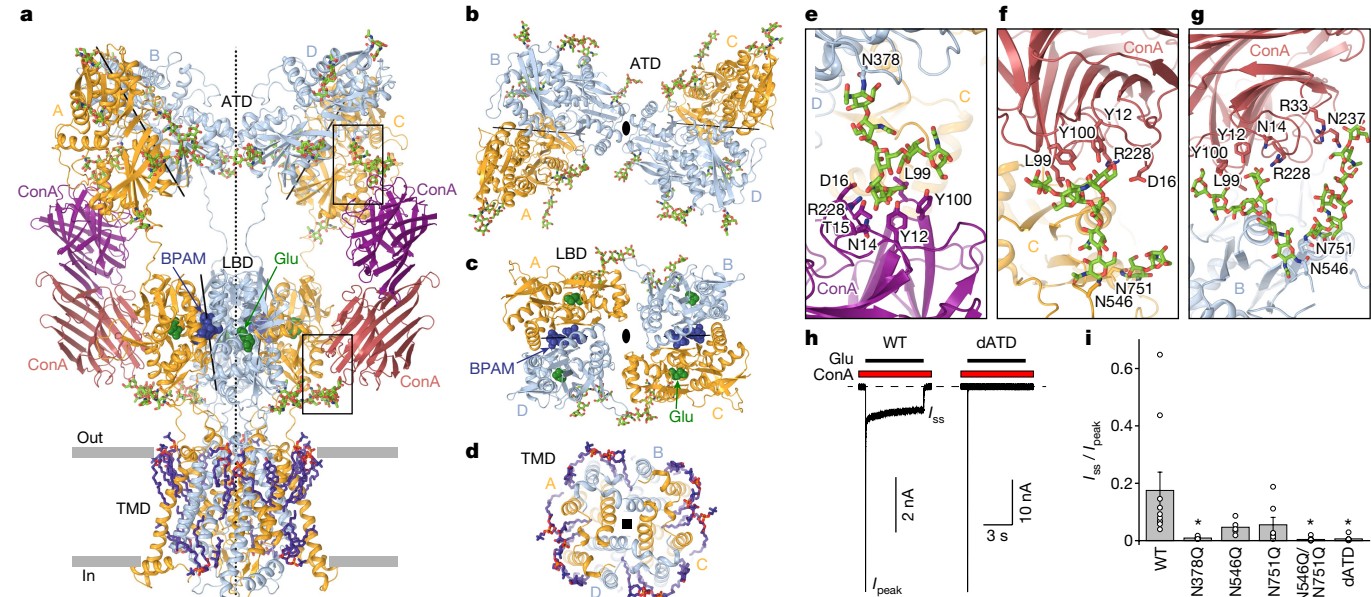

**Fig. 2 | The structure of GluK2$_{Glu-ConA-BPAM}$. a,** The GluK2$_{Glu-ConA-BPAM}$ structure viewed parallel to the membrane, with subunits A and C coloured orange, B and D in light blue, and ConA in purple and magenta. Glu (dark green) and BPAM (dark blue) are shown as space-filling models. Carbohydrates (light green) and lipids (violet) are shown as sticks. The axes of the two-fold overall symmetry and local symmetry of the ATD and LBD dimers are shown as dashed and solid black lines, respectively. Boxed regions are expanded in **e–g**. **b–d,** Extracellular views of ATD (**b**), LBD (**c**) and TMD (**d**) layers. **e–g,** ATD–ConA (**e**), LBD–ConA type I (**f**) and II (**g**) interfaces, with glycosylated asparagines and ConA residues in contact with carbohydrates shown as sticks. **h,** Whole-cell patch-clamp currents recorded at –60 mV membrane potential from HEK 293F cells expressing wild-type (WT) or dATD GluK2 in response to a 6-s application of 3 mM Glu in the continuous presence of 0.3 mg ml$^{-1}$ ConA. $I_{peak}$, peak current; $I_{ss}$, steady-state current. **i,** Steady-state to peak current ratio for wild-type and indicated mutant GluK2 channels. Data are mean ± s.e.m. Biologically independent measurements: $n = 10$ for wild type, $n = 6$ for N378Q, $n = 5$ for N546Q, $n = 7$ for N751Q, $n = 10$ for N546Q/N751Q and $n = 8$ for dATD. Data are representative of five independent experiments. Two-sided two-sample $t$-test, $*P < 0.05$. $P = 0.029$ for N378Q, $P = 0.078$ for N546Q, $P = 0.109$ for N751Q, $P = 0.026$ for N546Q/N751Q and $P = 0.027$ for dATD.

## Time-resolved cryo-EM

For time-resolved cryo-EM, we built a solenoid-driven plunger[43,44] that enabled freezing of GluK2 protein sample within 2–3 s after adding Glu in the presence of ConA and BPAM (Methods). In agreement with functional experiments (Fig. 1a,b), short exposure to Glu revealed an ensemble of KAR gating conformations. Processing of cryo-EM data revealed two main compositionally different groups of particles: with ConA bound and without ConA bound (Fig. 1c and Extended Data Figs. 1–3). The ConA-bound group of GluK2 particles includes two subgroups, with one or two ConA dimers bound. Both ConA-bound subgroups showed a distinct separation of the ATD and LBD layers, with ConA molecules serving as spacers (Extended Data Fig. 4a). Notably, the ATD and LBD layers in ConA-bound structures showed a dimer-of-dimers domain arrangement, typical for closed-state GluK2 and AMPAR structures[12,15,17,37]. Although the ATD layer and individual LBDs maintained similar conformations (Extended Data Fig. 4b–j), particles representing GluK2 with no ConA bound demonstrated different domain arrangement in the LBD layer (Fig. 1d and Extended Data Fig. 4a). Of these particles, 25% yielded a 3D reconstruction with a nearly four-fold symmetrical LBD layer, which resembled GluK2 structures in the presumed desensitized[12] and apo[18] states (Extended Data Fig. 5). Additionally, 22% of particles showed a mixed domain arrangement, with one intact LBD dimer and one LBD dimer separated into monomers.

## Structure of GluK2 bound to Glu, ConA and BPAM

The most compositionally complex GluK2–Glu–ConA–BPAM (GluK2$_{Glu-ConA-BPAM}$) structure has an overall two-fold rotational symmetry and two bound ConA dimers (Fig. 2a). The core of GluK2$_{Glu-ConA-BPAM}$

is a GluK2 homotetramer with the ATD and LBD layers assembled as dimers-of-dimers where A–B and C–D dimers of the ATD, as well as the A–D and B–C dimers of the LBD, are arranged symmetrically around the overall two-fold symmetry axis (Fig. 2b,c), with domain swapping between layers. Similar to the closed-state GluK2–BPAM (GluK2$_{BPAM}$) structure[17], the ATD and LBD dimers have individual local two-fold symmetry axes that do not coincide with the overall two-fold symmetry axis. Four TMDs are connected to the corresponding LBDs by S1–M1, M3–S2 and S2–M4 linkers and assemble a pseudo-four-fold symmetrical ion channel (Fig. 2d).

A distinct feature of GluK2$_{Glu-ConA-BPAM}$ compared to GluK2$_{BPAM}$ is a marked separation of the ATD and LBD layers, emphasized by an increase from 59 to 75 Å in the distance between their centres of mass (COMs) and lack of the ATD–LBD interfaces in the A and C subunits that are present in GluK2$_{BPAM}$ (Extended Data Fig. 4a). Instead, the ATD and LBD in the A and C subunits are connected through ConA dimers acting as extension spacers. One protomer in the ConA dimer contacts the ATD, whereas the second interacts with the LBD. The ATD–ConA contact is mediated by carbohydrates attached to N378, the glycosylation site conserved among GluK1–3 subunits (Fig. 2e and Supplementary Fig. 1). There were two types of ConA–LBD contacts (Extended Data Figs. 1 and 6), with the predominant type I contacts involving the A or C subunit LBD (Fig. 2f) and type II contact involving the B or D subunit LBD (Fig. 2g). The type I ConA–LBD contact is mediated by carbohydrates attached to N546, the glycosylation site conserved among GluK1–3 subunits, and the type II contact appears to also involve carbohydrates attached to N751, the glycosylation site conserved over all GluK subunits (Supplementary Fig. 1).

We mutated asparagine residues N378, N546 and N751 to glutamine and compared the effect of ConA on mutant and wild-type GluK2 using patch-clamp current recordings (Fig. 2h,i). The N378Q mutation

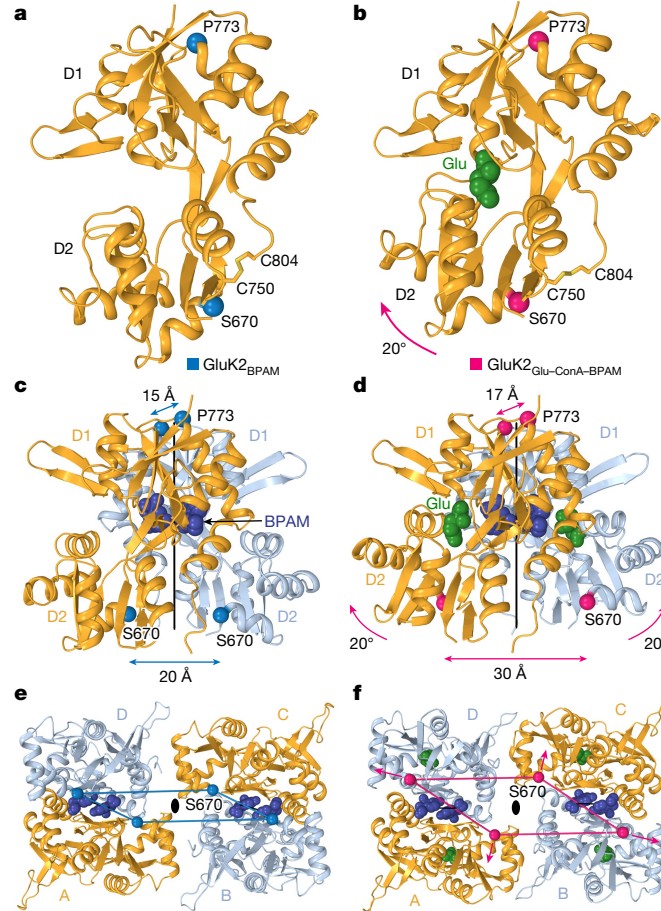

**Fig. 3 | Comparison of LBDs in GluK2<sub>BPAM</sub> and GluK2<sub>Glu–ConA–BPAM</sub>. a,b,** Single LBD in GluK2<sub>BPAM</sub> (**a**; Protein Data Bank (PDB) ID: 8FWS) and GluK2<sub>Glu–ConA–BPAM</sub> (**b**), with Glu shown as a space-filling model (green) and Cα atoms of S670 and P773 as blue and pink spheres, respectively. The 20° rotation of the lobe D2 towards D1 in GluK2<sub>Glu–ConA–BPAM</sub> upon Glu binding is illustrated by the pink arrow. **c,d,** LBD dimers in GluK2<sub>BPAM</sub> (**c**) and GluK2<sub>Glu–ConA–BPAM</sub> (**d**), with BPAM shown as a space-filling model (dark blue) and the axes of the local two-fold rotational symmetry as continuous black lines. Distances between Cα atoms of S670 and P773 are indicated. **e,f,** Intracellular view of LBD tetramers in GluK2<sub>BPAM</sub> (**e**) and GluK2<sub>Glu–ConA–BPAM</sub> (**f**). The Cα atoms of S670 are connected by blue (**e**) or pink (**f**) lines and pink arrows illustrate the expansion forces applied to the M3–S2 linkers that induce channel opening.

eliminated the steady-state current in response to Glu application in the presence of ConA, indicating the critical role of carbohydrates attached to N378 in PAM action. Although single N546Q and N751Q substitutions substantially reduced the steady-state currents, dramatic current reduction was only achieved in the double mutant N546Q/N751Q, suggesting the interchangeable capability of carbohydrates attached to N546 and N751 to serve as ConA contact points. Complete elimination of the ConA effect was also achieved by deletion of the entire ATD (dATD construct, which does not include N378). Our experiments with mutant GluK2 channels therefore confirmed glycosylation of residues N378, N546 and N751 to be critical for PAM action of lectin ConA on GluK2 receptors.

Despite observations of several configurations of the GluK2–ConA–Glu–BPAM complex, with one or two molecules of ConA bound and type I or type II ConA–LBD interfaces (Figs. 1c and 2e–g and Extended Data Fig. 6), all these configurations demonstrated a similar structure of the LBD–TMD region. We improved the resolution of this region by averaging the focused signal across all configurations and further refer to this LBD–TMD structure as GluK2<sub>Glu–ConA–BPAM</sub>.

## Activated LBDs

Compared with the agonist-unbound open-clamshell LBDs in the closed-state structure of GluK2<sub>BPAM</sub> (Fig. 3a), the individual LBDs in GluK2<sub>Glu–ConA–BPAM</sub> show Glu molecules bound (Extended Data Fig. 3a), with each clamshell closed via a 20° rotation of the lower lobe D2 towards the upper lobe D1 (Fig. 3b). In the context of back-to-back LBD dimers that maintain their two-fold rotational symmetry and intact D1–D1 interfaces in both GluK2<sub>BPAM</sub> and GluK2<sub>Glu–ConA–BPAM</sub>, the 20° closures of individual LBDs lead to a marked separation of the D2 lobes, exemplified by an increase in the distance between Cα atoms of S670 from 20 Å in GluK2<sub>BPAM</sub> to 30 Å in GluK2<sub>Glu–ConA–BPAM</sub> (Fig. 3c,d). In the context of the dimer-of-dimers arrangement, the separation of the D2 lobes results in LBD tetramer expansion (Fig. 3e,f). A similar expansion of the LBD tetramer in AMPARs leads to ion channel opening[38,39]. Therefore, we assessed the conformation of the GluK2<sub>Glu–ConA–BPAM</sub> ion channel pore.

## Open channel

We measured the channel pore radius in GluK2<sub>Glu–ConA–BPAM</sub> and observed that the gate region became much wider than in the previously published GluK2<sub>BPAM</sub> structure (Fig. 4a,b). The apparent opening of the extracellular pore (Fig. 4c,d) is comparable to or larger than the one observed in open-state AMPARs[45,46] (Extended Data Fig. 7). Given the highly conserved residues lining the gate region of KARs and AMPARs, specifically the signature ionotropic glutamate receptor SYTANLAAF motif[47], the pore of GluK2<sub>Glu–ConA–BPAM</sub> is also expected to be open. Indeed, during equilibrium molecular dynamics simulations of GluK2<sub>Glu–ConA–BPAM</sub> incorporated into a lipid bilayer and surrounded by aqueous electrolyte solution, the channel pore remained continuously hydrated, and monovalent cations freely permeated through the gate region in both directions in the absence of applied voltage (Extended Data Fig. 8 and Supplementary Video 1). By contrast, no water hydration or cation permeation was observed for the gate region during molecular dynamics simulations of GluK2<sub>BPAM</sub> (Extended Data Fig. 8).

Comparison of the gate region in GluK2<sub>Glu–ConA–BPAM</sub> and GluK2<sub>BPAM</sub> reveals the key molecular changes associated with channel opening (Fig. 4e,f). In response to Glu binding, the separation of D2 lobes in LBD dimers leads to LBD layer expansion (Fig. 3c–f), characterized by the substantial increase in the distance between D2 lobes of diagonal B and D subunit LBDs (Fig. 4f). The LBD layer expansion increases tension in the M3–S2 linkers that in turn leads to kinking of the M3 helices at a gating hinge around L655 (Fig. 4e,f). Of note, the M3 kinking associated with channel opening occurs in both pairs of diagonal subunits, A–C and B–D, notably distinct from AMPARs in which the M3 kinking was experimentally observed in subunits B and D only and at a different location, two residues towards the N terminus compared with KAR[38,46] (Extended Data Fig. 7d,e). While molecular dynamics simulations predicted kinking of all four M3 helices in AMPARs as well[46], it remains to be confirmed experimentally.

## Glutamate-only bound structures

Particles representing GluK2 with no ConA bound (Fig. 1c and Extended Data Fig. 4a) yielded two structures, GluK2<sub>Glu-4fold</sub>, which has a nearly four-fold symmetrical LBD layer (Fig. 5a), and an apparently asymmetric structure, GluK2<sub>Glu-asym</sub> (Fig. 5b). Both structures revealed Glu but not BPAM molecules bound to the LBDs, despite incubation of the sample with saturating BPAM concentration. The apparent reason for the lack of BPAM bound to GluK2<sub>Glu-4fold</sub> is the absence of LBD dimer interfaces that harbour BPAM molecules in GluK2<sub>BPAM</sub> and GluK2<sub>Glu–ConA–BPAM</sub> (Figs. 2 and 3). Indeed, dissociation of LBD dimers in GluK2<sub>Glu-4fold</sub> occurs owing to an approximately 100° rotation of the B and D subunit LBDs (Fig. 5c), which has previously been observed in the presumed desensitized[12] and apo[18] state GluK2 structures (Extended Data Fig. 5).

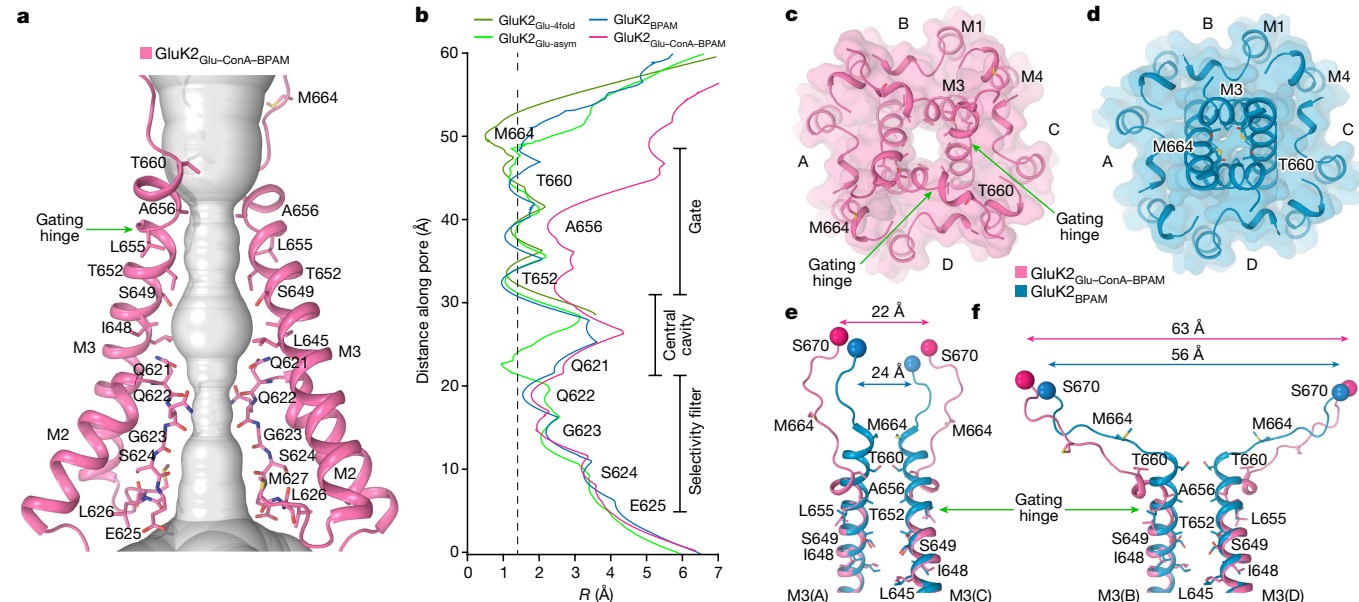

**Fig. 4 | Open pore in GluK2_{Glu-ConA-BPAM}.** **a**, Pore-forming segments M2 and M3 in GluK2_{Glu-ConA-BPAM}, with residues lining the pore shown as sticks. Only two subunits (A and C) of the four are shown, with the front and back subunits (B and D) omitted for clarity. The pore profile is shown as a space-filling model (grey). The green arrow indicates the gating hinge where M3 bends during channel opening. **b**, Pore radius for GluK2_{BPAM} (blue), GluK2_{Glu-ConA-BPAM} (pink), GluK2_{Glu-4fold} (dirty green) and GluK2_{Glu-asym} (green) calculated using HOLE. The vertical dashed line denotes the radius of a water molecule (1.4 Å). **c**,**d**, Extracellular view of the ion channel semi-transparent surface in GluK2_{Glu-ConA-BPAM} (**c**) and GluK2_{BPAM} (**d**; PDB ID: 8FWS). **e**,**f**, Superposition of the pore-lining segments M3 and M3–S2 linkers in subunits A and C (**e**) and B and D (**f**), with residues lining the pore in the gate region shown as sticks. Distances between Cα atoms of S670 are indicated.

Similar to GluK2_{Glu-4fold}, the B–C subunit LBD dimer is dissociated in GluK2_{Glu-asym} owing to a rotation of approximately 100° of the subunit B LBD (Fig. 5d). Although the A–D subunit LBD dimer in GluK2_{Glu-asym} is preserved, the interface between protomers is severely altered compared with the GluK2_{BPAM} and GluK2_{Glu-ConA-BPAM} structures. Indeed, the D1–D1 interface in the A–D LBD dimer of GluK2_{Glu-asym} undergoes rupture, resulting in marked separation of the D1 lobes (Fig. 5e) compared with GluK2_{BPAM} (Fig. 5f) and GluK2_{Glu-ConA-BPAM} (Fig. 5g). This rupture of the interface that harbours BPAM molecules in GluK2_{BPAM} and GluK2_{Glu-ConA-BPAM} distorts BPAM binding sites and explains why this PAM is not bound to the GluK2_{Glu-asym} A–D LBD dimer. Of note, during the D1–D1 interface rupture, the GluK2_{Glu-asym} A–D LBD dimer loses its two-fold rotational symmetry, a defining aspect of GluK2_{BPAM} and GluK2_{Glu-ConA-BPAM} structures. Signified by the appearance of a side cleft (Fig. 5e), this loss of two-fold symmetry, together with the separation of D1 lobes, represents characteristic features of the desensitized-state dimer in the complex of AMPAR with its auxiliary subunit GSG1L[48,49] (Extended Data Fig. 9). Notably, although the D1 lobe separation characterizes desensitized AMPAR complexes with all transmembrane AMPAR regulatory proteins (TARPs), their two-fold symmetry appears to be preserved in complexes with γ2 and γ5 (Extended Data Fig. 9c), but not in complexes with γ8 subunits[39,49,50].

The pores in Glu-bound GluK2_{Glu-4fold} and GluK2_{Glu-asym} appear to be non-conducting (Figs. 4b and 6a,b), consistent with these being desensitized ionotropic glutamate receptors. These agonist-bound structures cannot represent an open state because Glu binding does not cause the LBD layer expansion (Fig. 3f) that is required to apply tension to the LBD–TMD linkers and drive pore opening. Instead, an approximately 100° rotation of B and D LBDs reduces tension in the linkers by bringing their attachment points closer together. Indeed, among altered distances between attachment points of the diagonal M3–S2 linkers, the most critical for driving pore opening distances between the diagonal subunits B and D undergo substantial reduction in both GluK2_{Glu-4fold} (Fig. 6c) and GluK2_{Glu-asym} (Fig. 6d), the opposite of what happens in the open-state GluK2_{Glu-ConA-BPAM} structure (Fig. 4f).

## Discussion

We utilized two PAMs—ConA and BPAM—to capture the open conformation of the GluK2 KAR channel activated by Glu. Activation is driven by Glu-induced closure of individual LBD clamshells (Fig. 3a,b), which is converted into separation of D2 lobes in back-to-back LBD dimers (Fig. 3c,d) and expansion of the LBD tetramer (Fig. 3e,f). During activation, ConA and BPAM appear to work synergistically to delay desensitization (Fig. 1a,b). Although BPAM alone acting as a glue is too weak to substantially impede the breakdown of the D1–D1 LBD interface associated with desensitization, ConA bound to the LBD D2 lobes seems to sterically hinder this breakdown further. In addition, the role of ConA as a spacer between the ATD and LBD layers appears to be an elimination of ATD–LBD interfaces in subunits A and C (Extended Data Fig. 4a) that might restrain the activation-associated expansion of the LBD tetramer. Applied to the LBD–TMD linkers (Fig. 4e,f), the expansion of the LBD tetramer causes opening of the ion channel pore (Fig. 4a–d and Extended Data Fig. 8).

KAR activation appears to resemble that of AMPARs, and probably occurs through a pre-active state, in which Glu has already bound but the channel is not yet open[35,38,39,51-53]. However, opening of the GluK2 channel involves kinking of all four M3 helices (Fig. 4e,f), whereas all available open-state structures of AMPARs have M3 kinked in subunits B and D only (Extended Data Fig. 7d,e). One possible explanation is that among open states with different ion conductance (states O1–O4, with increasing channel opening), GluK2_{Glu-ConA-BPAM} represents the maximal conductance O4 state, and AMPAR structures represent intermediate conductance O1 and O2 states[46]. Alternatively, LBD–TMD linkers in AMPARs and KARs that have distinct amino acid sequences may be engaged in specific interactions, which direct the LBD layer expansion forces to the M3 segments in a distinct manner. Distinct directionality of applied forces is suggested by a rotation of approximately 12° of the LBD layer in the open-state structure of GluK2 compared with GluA2 (Extended Data Fig. 7a–c). Since all available open-state structures of AMPARs have been solved in complex with auxiliary subunits, the

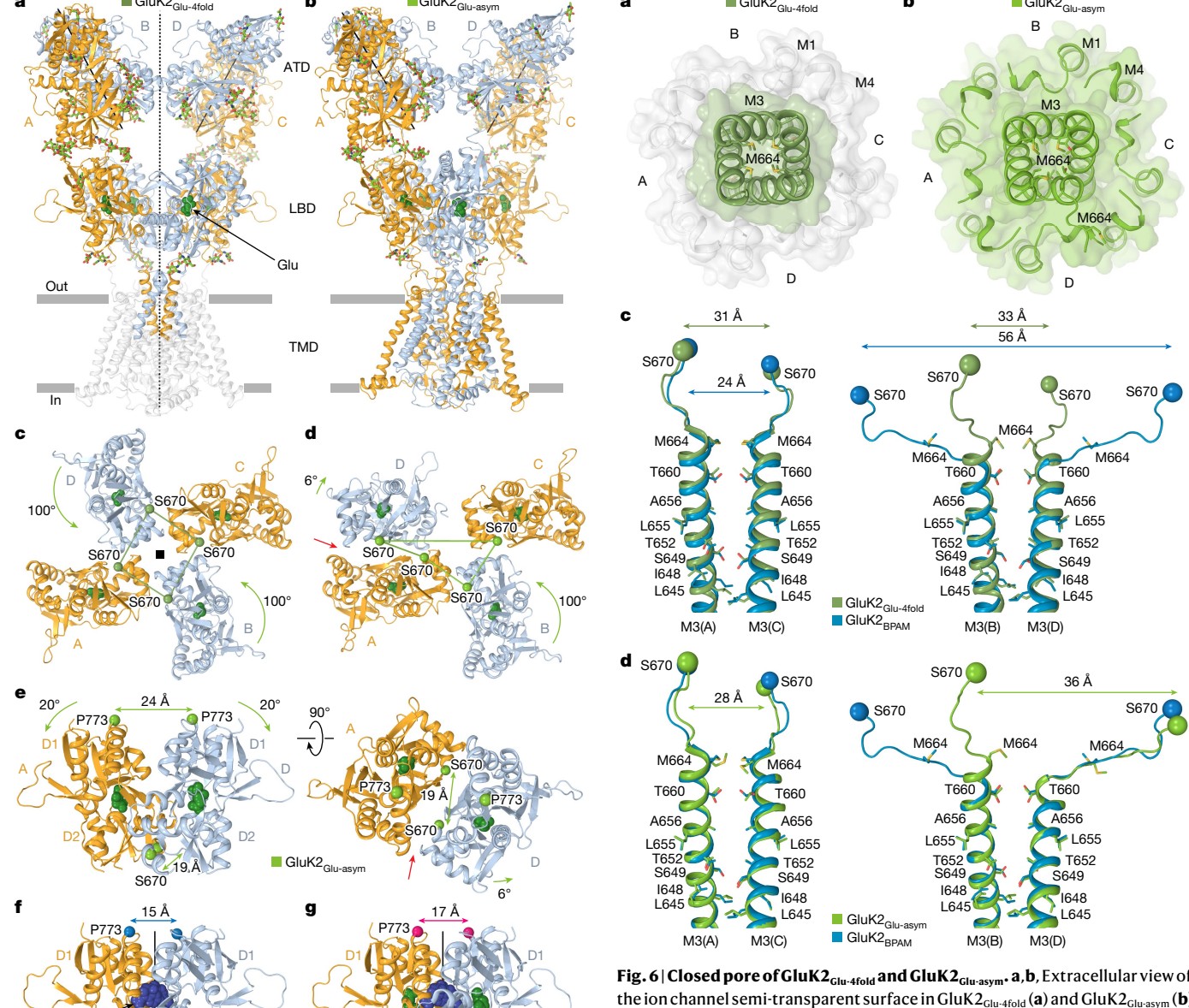

**Fig. 5 | Structures of GluK2$_{Glu-4fold}$ and GluK2$_{Glu-asym}$. a**,**b**, Structures of GluK2$_{Glu-4fold}$ (**a**) and GluK2$_{Glu-asym}$ (**b**) viewed parallel to membrane, with subunits A and C coloured orange and B and D coloured light blue. Glu (dark green) is shown as space-filling models and carbohydrates (light green) are shown as sticks. The axis of the overall two-fold symmetry in GluK2$_{Glu-4fold}$ is shown as a dashed line, and axes of the local two-fold symmetry for ATD dimers are shown as solid lines. **c**,**d**, Intracellular view of the LBD layer in GluK2$_{Glu-4fold}$ (**c**) and GluK2$_{Glu-asym}$ (**d**), with the Cα atoms of S670 (green spheres) connected by lines and green arrows illustrating the rotation of subunit B and D LBDs relative to their positions in GluK2$_{BPAM}$. The red arrow points to the cleft between the GluK2$_{Glu-asym}$ protomers, indicating the loss of the LBD dimer two-fold rotational symmetry. **e**, Side (left) and top (right) views of the LBD dimer in GluK2$_{Glu-asym}$, with distances between Cα atoms of S670 and P773 indicated. Green arrows illustrate the 20° rotation of the lobes D1 towards D2 upon Glu binding compared to GluK2$_{BPAM}$. **f**,**g**, Similar side view of LBD dimers in GluK2$_{BPAM}$ (**f**) and GluK2$_{Glu-ConA-BPAM}$ (**g**), with BPAM shown as a space-filling model (dark blue) and the axes of the local two-fold rotational symmetry shown as solid black lines.

**Fig. 6 | Closed pore of GluK2$_{Glu-4fold}$ and GluK2$_{Glu-asym}$. a**,**b**, Extracellular view of the ion channel semi-transparent surface in GluK2$_{Glu-4fold}$ (**a**) and GluK2$_{Glu-asym}$ (**b**). **c**,**d**, Superposition of the pore-lining segments M3 and M3–S2 linkers in subunits A and C (left) and B and D (right) of GluK2$_{BPAM}$ (PDB ID: 8FWS, blue) and GluK2$_{Glu-4fold}$ (**c**; dirty green) or GluK2$_{Glu-asym}$ (**d**; green), with residues lining the pore in the gate region shown as sticks. Distances between Cα atoms of S670 are indicated.

difference in M3 behaviour during channel opening might also be due to interactions with auxiliary subunits.

The states represented by the GluK2$_{Glu-4fold}$ and GluK2$_{Glu-asym}$ structures remain unresolved. These structures might represent different stages of desensitization, with GluK2$_{Glu-asym}$ being an intermediate state. This would be consistent with previous studies in which a state identical to GluK2$_{Glu-4fold}$ was declared the main desensitized state[12]. In this scenario, desensitization would begin similarly to that of AMPARs, from the pre-active state, with rupture of D1–D1 interfaces in both LBD dimers. In contrast to AMPARs however, desensitization in KARs would proceed with further rupture of one LBD dimer (GluK2$_{Glu-asym}$), followed by the second (GluK2$_{Glu-4fold}$) through rotation of about 100° of B and D LBDs. A missing intermediate desensitized state (GluK2$_{Glu-2fold}$) that was not revealed in this study is expected to have two LBD dimers with ruptured D1–D1 interfaces. However, one inconsistency of this model is that the terminal desensitized-state GluK2$_{Glu-4fold}$ is identical to the recently determined apo conformation[18] (Extended Data Fig. 5). Moreover,

although it is conceivable that forces generated by Glu-induced LBD closure can be directed to rotate the B and D LBDs by 100°, it remains unclear what, aside from diffusion, could possibly cause such rotations in the reverse direction during recovery from desensitization.

An alternative explanation for the observed structural behaviour is that GluK2$_{Glu-4fold}$ represents a structural artefact, with LBD dimers dissociating into monomers owing to specific forces applied to KARs associated, for instance, with the air–liquid interface of the cryo-EM grid. According to this alternative, the putative GluK2$_{Glu-2fold}$ structure with two LBD dimers, each having ruptured D1–D1 interface, would be the ultimate desensitized state, analogous to AMPARs. Accordingly, the GluK2$_{Glu-asym}$ structure would then represent a semi-broken state, in which one LBD dimer has a natural desensitized conformation, and another is dissociated into monomers. These questions remain to be addressed in future studies.

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

# Methods

## Plasmids and strains

DNA for the full-length rat GluK2 (GenBank CAA77778.1), with V at position 567, C at position 571 and Q at position 621 (the Q/R site), was amplified using PCR and subsequently inserted into the pEG BacMam vector[54]. This vector included a thrombin cleavage site (LVPRG) at the C-terminus, eGFP, and streptavidin affinity tag (WSHPQFEK). The Bacmid and baculoviruses were produced in Sf9 cells (GIBCO) cultured in Sf-900 III SFM media (GIBCO) at 27 °C using standard methods[54]. Recombinant rat GluK2 was expressed in HEK 293F cells (ATCC) cultured in Freestyle 293 expression medium (GIBCO) at 30 °C and 6% $CO_2$. For patch-clamp experiments, HEK 293F cells were employed and maintained at 37 °C and 6% $CO_2$ in Dulbecco's Modified Eagle's Medium, supplemented with 10% fetal bovine serum.

## Protein expression and purification

Purification of recombinant GluK2 followed previously established procedures with slight modifications[17]. In brief, P2 viruses produced in Sf9 cells were introduced to HEK 293F cells (Gibco, R79007) and incubated at 37 °C with 6% $CO_2$. After 12–15 h post-transduction, 10 mM sodium butyrate was added to the cells, and the temperature was then lowered to 30 °C. The cultured cells were harvested 72 h post-transduction through low-speed centrifugation (5,500$g$, 10 min), followed by a wash with 1× PBS pH 7.4 and another centrifugation (5,500$g$, 15 min). The resulting cell pellet was resuspended in 50 ml of ice-cold lysis buffer composed of 150 mM NaCl, 20 mM Tris-HCl pH 8.0, 1 mM β-mercaptoethanol, 0.8 µM aprotinin, 2 µg ml$^{-1}$ leupeptin, 2 µM pepstatin A, and 1 mM phenylmethylsulfonyl fluoride (PMSF). The cells were lysed by sonication for 3 min (15 s on and 15 s off). The cell lysate was centrifuged (9,900$g$, 15 min) to remove cell debris, and the resulting supernatant was subjected to ultracentrifugation (186,000$g$, 1 h) to isolate cell membranes. The obtained cell membranes were mechanically homogenized and solubilized for 2–3 h in a buffer consisting of 150 mM NaCl, 20 mM Tris-HCl pH 8.0, 1 mM β-mercaptoethanol, and 1% digitonin (Cayman Chemical Company, 14952). The insoluble material was removed by another round of ultracentrifugation (186,000$g$, 1 h). The supernatant was combined with pre-equilibrated streptavidin-linked resin (2 ml) and rotated for 10–14 h at 4 °C. Subsequently, the protein-bound resin was washed with 30–35 ml buffer containing 150 mM NaCl, 20 mM Tris-HCl pH 8.0, and 0.05% digitonin. The protein was eluted in 12–15 ml of the same buffer supplemented with 2.5 mM desthiobiotin. The eluted protein was concentrated and subjected to thrombin digestion (1:200 w/w) at 22 °C for 90 min to remove eGFP and the streptavidin affinity tag. The digest reaction was then injected into a Superose 6 10/300 GL size-exclusion chromatography column (GE Healthcare) pre-equilibrated with a buffer containing 150 mM NaCl, 20 mM Tris-HCl pH 8.0, and 0.05% digitonin. The peak fractions corresponding to tetrameric GluK2 were pooled, concentrated to 6–6.5 mg ml$^{-1}$ (~60 µM), and used for cryo-EM sample preparation. All procedures, unless otherwise specified, were carried out at 4 °C.

## Electron microscopy sample preparation and data collection

We utilized an in-house vitrification system developed for rapid grid preparation (Extended Data Fig. 10). This system was built using similar operational principles as the previously described vitrification devices[43,44] with several modifications. We used Kuhnke HD8286-R-F-24VDC and T Tulead 12 V DC as plunging and blotting solenoids, respectively. Solenoid holders, a magnetic lock, a blotting paper holder, and tweezer holders were designed in Blender 3.1.2 using the previously designed parts[43,44] as templates and printed using a Form 3 (Formlabs) 3D printer. All parts that we used to assemble a vitrification system stand on optical table were purchased from ThorLabs. Solenoid driver circuit boards were designed and assembled in-house. Solenoids were powered by 60 V (plunging) and 12 V (blotting) direct current power supply sources. The performance of blotting and plunging solenoids was controlled by an Axon Digidata 1440 A digitizer operated by the Clampex 10.7 software.

For cryo-EM grid preparation, we employed UltrAuFoil R 1.2/1.3, 300-mesh gold grids (EMS). Before sample application, the grids were treated in a PELCO easyGlow cleaning system (Ted Pella, 25 s, 15 mA) to render their surface hydrophilic. Purified GluK2 protein at a concentration of 60 µM was incubated with 500 µM BPAM344 for 20–30 min on ice. Subsequently, 90 µM of succinylated concanavalin A was rapidly added to the GluK2–BPAM mixture, and a 1.5 µl protein sample was promptly applied to both sides of the grid. Following this, 0.5 µl L-glutamate at 10 mM was applied to one side of the grid, blotted for 1.5 s from the other side, and immediately vitrified in liquid ethane cooled by liquid nitrogen. The total time between Glu application and plunge-freezing was less than 3 s. Starting from the addition of ConA to the GluK2–BPAM mix until vitrification, the entire process was completed in 40–45 s. The grids were imaged using a Titan Krios transmission electron microscope (Thermo Fisher Scientific) operating at 300 kV, equipped with a post-column GIF Quantum energy filter (slit width set to 20 eV) and a Gatan K3 direct electron detection camera. The total number of 22,990 images were collected in the counting mode across the defocus range of −1.0 to −2.0 µm and the pixel size of 0.83 Å. The total dose of ~58 e$^-$Å$^{-2}$ was attained by using the dose rate of ~16 e$^-$pixel$^{-1}$s$^{-1}$ during the 2.5-s exposure time across 50 frames.

## Image processing and three-dimensional reconstructions

Data were processed in cryoSPARC v4.4.1[55] (Extended Data Fig. 1). Movie frames were aligned using Patch Motion Correction algorithm. Contrast transfer function (CTF) estimation was performed using the patch CTF estimation. Following CTF estimation, micrographs were manually inspected and those with outliers in defocus values, ice thickness, and astigmatism as well as micrographs with lower predicted CTF-correlated resolution were excluded from further processing (individually assessed for each parameter relative to the overall distribution). Approximately 5,000 micrographs were used for blob picking and particle extraction, followed by multiple rounds of 2D classification to create templates for Template picking and training Topaz model[56]. After removal of duplicates, the total number of 8,660,229 particles were picked from the full set of micrographs using results from Template picker and Topaz and extracted with the 416-pixel box size and then binned to the 128-pixel box size. After several rounds of reference-free 2D classifications and heterogeneous refinements with one reference class created using ab initio reconstruction and three automatically generated 'garbage' classes, the best set of 746,323 particles was subjected to heterogeneous refinement into four classes. At this moment, we observed two classes with ConA bound (419,597 particles) and two classes without ConA density (326,726 particles). ConA-bound and ConA-unbound classes were separately subjected to the reference-based motion correction, and the resulting pool of particles with a 416-pixel box size was down sampled to a 256-pixel box size (or 1.34 Å pixel$^{-1}$) and used throughout the remaining processing steps.

Particles representing GluK2 without ConA were subjected to non-uniform refinement without applied symmetry restrains ($C_1$) and classified into 3 classes using a reference-free 3D classification algorithm. Particles from the best two classes underwent refinement using non-uniform refinement. This process included refinement with either $C_2$ symmetry (80,612 particles) imposed, resulting in a 3.36-Å reconstruction (GluK2$_{Glu-4fold}$) and a nearly four-fold symmetrical LBD layer, or without imposed symmetry ($C_1$, 72,375 particles), resulting in a 3.84-Å reconstruction (GluK2$_{Glu-asym}$) and an apparently asymmetric structure.

Particles representing GluK2 with ConA bound were subjected to the non-uniform refinement without application of symmetry restraints ($C_1$). Subsequently, focused masks around the ATD, ConA and LBD–TMD were created and used for the local refinements of the corresponding

regions. Local refinements with the focused masks resulted in 3.50-Å reconstruction ($C_2$) of the ATD region, 3.69-Å reconstruction ($C_1$) of the ConA dimer, and 3.40-Å reconstruction ($C_1$) of the LBD–TMD region.

Additionally, heterogeneous refinement was used to classify all ConA-bound particles into five classes, yielding GluK2 classes with distinct modes of ConA binding to a full-length GluK2 tetramer: three classes with two ConA dimers bound and two classes with a single ConA dimer bound. Of these five classes, two most populated were subjected to non-uniform refinement, resulting in a 6.66-Å reconstruction ($C_1$, 77,587 particles) with a single ConA dimer bound, and a 4.29-Å reconstruction ($C_2$, 109,827 particles) with two ConA dimers bound.

Subsequently, the combine_focused_maps algorithm implemented in Phenix was used to create composite maps for GluK2 bound to either one or two ConA dimers, with 4.29-Å and 6.66-Å resolutions, as well as the locally refined ATD, ConA and LBD–TMD regions as inputs. The reported resolution for the final maps was estimated in cryoSPARC using the gold standard Fourier shell correlation of the Fourier shell correlation (FSC) = 0.143 criterion. The local resolution was calculated in cryoSPARC using the FSC = 0.5 criterion. Cryo-EM densities were visualized using UCSF ChimeraX[57]. Structural biology applications used in this project adhered to and were configured by SBGrid.

## Model building, refinement and validation

Initially, the GluK2–ConA model was constructed in Coot[58] using cryo-EM density alongside the coordinates from the full-length cryo-EM structure of GluK2 (PDB ID: 8FWQ) and the X-ray crystal structure of ConA (PDB ID: 3ENR) as references. The models of GluK2 and ConA were docked into the GluK2–ConA map in Chimera[59]. Visual inspection ensured a general fit, and manual model building was carried out in Coot. The resolution and quality of the maps facilitated the unambiguous completion of model building. To check for overfitting, the models were subjected to a coordinate shift of 0.5 Å (using shake) in Phenix[60]. Each shaken model was refined against a corresponding unfiltered half map, and densities were generated from the resulting models in Chimera. The resultant models underwent real-space refinement in Phenix and were visualized in Chimera or Pymol (The PyMOL Molecular Graphics System, version 2.0, Schrödinger). Validation statistics for the models are summarized in Extended Data Table 1. Domain rotations were determined using the DynDom server (http://dyndom.cmp.uea.ac.uk/dyndom/). The pore radius was calculated using HOLE[61].

## Patch-clamp recordings

DNA encoding GluK2 (described in 'Plasmids and strains') was introduced into a pIRES plasmid for expression in eukaryotic cells that were engineered to produce green fluorescent protein via a downstream internal ribosome entry site[53]. HEK 293 cells (ATCC, CRL-1573) grown on glass coverslips in 35-mm dishes were transiently transfected with 1–5 µg plasmid DNA using Lipofectamine 2000 Reagent (Invitrogen). Recordings were made 24 to 48 h after transfection at room temperature. Currents from whole cells, typically held at –60 mV potential, were recorded using Axopatch 200B amplifier (Molecular Devices), filtered at 5 kHz, and digitized at 10 kHz using low-noise data acquisition system Digidata 1440 A and pCLAMP 10.2 software (Molecular Devices). The external solution contained (in mM): 150 NaCl, 2.4 KCl, 4 CaCl$_2$, 4 MgCl$_2$, and 10 HEPES pH 7.3; 7 mM NaCl was added to the extracellular activating solution containing 3 mM Glu to improve visualization of the border between two solutions coming out of a two-barrel theta glass pipette, which allowed its more precise positional adjustment for faster solution exchange. The internal solution contained (in mM): 150 CsF, 10 NaCl, 10 EGTA, 20 HEPES pH 7.3. Rapid solution exchange was achieved with a two-barrel theta glass pipette controlled by a piezo-electric translator. Typical 10–90% rise times were 200–300 µs, as measured from junction potentials at the open tip of the patch pipette after recordings. Data analysis was performed using the Clampfit 10.3 and Origin 2023 software (OriginLab Corporation).

## System preparation for molecular dynamics simulations

The structures of GluK2$_{BPAM}$ (PDB ID: 8FWS) and GluK2$_{Glu–ConA–BPAM}$ were used as initial atomic coordinates for molecular dynamics simulations. GluK2$_{BPAM}$ structure comprised the LBD, TMD and linkers between them and included residues R431 to A850. Similarly, the GluK2$_{Glu–ConA–BPAM}$ structure used in molecular dynamics simulations included residues starting from S429 in the LBD and ending with R874 in the TMD. For the molecular dynamics system of GluK2$_{BPAM}$, we kept the protein, Na$^+$ ions and BPAM molecules from the cryo-EM structure and removed all other molecules. For the GluK2$_{Glu–ConA–BPAM}$ system, we kept the protein, BPAM, Glu, and cholesterol molecules from the cryo-EM structure and removed all other molecules. Each simulation box was constructed in CHARMM-GUI membrane builder[62,63] by inserting the protein into a POPC bilayer and solvating it with TIP3P water molecules and 150 mM KCl. The systems were set up for molecular dynamics simulations using the "tleap" module of the AmberTools20 package[64]. Parametrization of all the ligands was carried out using the general AMBER force field (GAFF2)[65]. The total number of atoms in the final simulation boxes was 304,680 for the GluK2$_{BPAM}$ system and 313,937 for the GluK2$_{Glu–ConA–BPAM}$ system. The boxes contained 69,896 water molecules for GluK2$_{BPAM}$ and 72,667 for GluK2$_{Glu–ConA–BPAM}$. The GluK2$_{BPAM}$ system contained 6 Na$^+$, 207 K$^+$ and 189 Cl$^-$ ions, while the GluK2$_{Glu–ConA–BPAM}$ system contained 212 K$^+$ and 196 Cl$^-$ ions. The number of lipid molecules was 507 for GluK2$_{BPAM}$ and 504 for GluK2$_{Glu–ConA–BPAM}$.

## Molecular dynamics simulation protocols

The Pmemd.cuda program of the Amber20 molecular dynamics software package was used for all molecular dynamics simulations[64]. Amber FF99SB–ILDN[66] force field parameters were used for protein and ions, TIP3P model for water, and Lipid14 force field[67] parameters for lipids. All equilibration and production simulations were performed in NPT ensemble at 300 K temperature and 1 bar pressure with anisotropic pressure scaling. The temperature was controlled using Langevin thermostat with a collision frequency of 1 ps$^{-1}$ and the pressure was controlled using Berendsen barostat with a relaxation time of 1 ps as implemented in Amber20. All covalent bonds involving hydrogen atoms were constrained using the SHAKE algorithm[68], with the integration time step of 2 fs. The long-range electrostatic interactions were approximated using the particle mesh Ewald method[69], with a non-bonded interaction cutoff radius of 10 Å. Periodic boundary conditions were applied in all directions.

Each system was minimized while keeping restraints on protein Cα and ligand heavy atoms. Next, water and ions were equilibrated at constant volume molecular dynamics simulations as the temperature was gradually increased from 0 to 300 K, with all protein, ligand and lipid heavy atoms harmonically restrained at their energy minimized positions with the force constant of 40 kcal mol$^{-1}$ Å$^{-2}$. The systems were then equilibrated for 100 ns at constant pressure, gradually releasing the restraints on the protein and ligands to 0.5 kcal mol$^{-1}$ Å$^{-2}$. Production simulations were carried out for 200 ns without any restraints on the pore-forming helixes or non-helical structures.

## Molecular dynamics trajectory analysis

Post-processing and analysis of the trajectories were carried out using CPPTRAJ[70] module of AmberTools20 and VMD 1.9.4[71]. VMD 1.9.4 was used to visualize trajectories and generate simulation video, and PyMOL was used to generate simulation snapshot figures.

## Statistics and reproducibility

Statistical analysis was performed using Origin 2023. Statistical significance was calculated using two-sample $t$-test, with the significance assumed if $P < 0.05$. In all figure legends, $n$ represents the number of independent biological replicates. All quantitative data were presented as mean ± s.e.m.

## Reporting summary

Further information on research design is available in the Nature Portfolio Reporting Summary linked to this article.

## Data availability

The cryo-EM density maps have been deposited to the Electron Microscopy Data Bank (EMDB) under the accession codes EMD-44129 (GluK2$_{Glu-2\times ConA-BPAM}$, composite map), EMD-44130 (GluK2$_{Glu-1\times ConA-BPAM}$, composite map), EMD-44128 (GluK2$_{Glu-ConA-BPAM}$, LBD–TMD), EMD-44131 (GluK2$_{Glu-4fold}$), EMD-44132 (GluK2$_{Glu-asym}$), EMD-44125 (ConA, type I), EMD-44124 (ConA, type II), EMD-44123 (GluK2$_{Glu-ConA-BPAM}$, ATD), EMD-44126 (GluK2$_{Glu-2\times ConA-BPAM}$, reference map) and EMD-44127 (GluK2$_{Glu-1\times ConA-BPAM}$, reference map). The atomic coordinates have been deposited to the Protein Data Bank (PDB) under the accession codes 9B36 (GluK2$_{Glu-2\times ConA-BPAM}$), 9B37 (GluK2$_{Glu-1\times ConA-BPAM}$), 9B35 (GluK2$_{Glu-ConA-BPAM}$, LBD–TMD), 9B38 (GluK2$_{Glu-4fold}$), 9B39 (GluK2$_{Glu-asym}$), 9B34 (ConA, type I) and 9B33 (ConA, type II). Source data are provided with this paper.

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

**Acknowledgements** The authors thank R. Grassucci and Z. Zhang for help with microscope operation and data collection; and I. A. Talyzina for assistance with building the in-house vitrification system. Some of this work was performed at the Columbia University Cryo-Electron Microscopy Center. A portion of this research was supported by NIH grant U24GM129547 and performed at the PNCC at OHSU and accessed through EMSL (grid.436923.9), a DOE Office of Science User Facility sponsored by the Office of Biological and Environmental Research. Some of this work was performed at the National Center for CryoEM Access and Training (NCCAT) and the Simons Electron Microscopy Center located at the New York Structural Biology Center, supported by the NIH Common Fund Transformative High Resolution Cryo-Electron Microscopy programme (U24 GM129539) and by grants from the Simons Foundation (SF349247) and NY State Assembly Majority. Some of this work was performed at the Stanford-SLAC Cryo-EM Center (S2C2), supported by the National Institutes of Health Common Fund Transformative High Resolution Cryo-Electron Microscopy programme (U24 GM129541). A.I.S. was supported by the National Institutes of Health grants (NS083660, NS107253, AR078814, CA206573).

**Author contributions** S.P.G., M.V.Y. and A.I.S. conceptualized the project. S.P.G., M.V.Y., K.D.N. and A.I.S. designed the experiments. S.P.G., M.V.Y. and T.P.N. made the constructs for protein expression and electrophysiology. S.P.G. and L.Y.Y. performed protein expression and purification. K.D.N. designed and assembled the in-house vitrification system. S.P.G. and K.D.N. made the grids. S.P.G., K.D.N. and L.Y.Y. collected the data. K.D.N. and S.P.G. processed the cryo-EM data. M.V.Y. performed patch-clamp recordings and electrophysiological data analysis. M.A. and M.G.K. designed computational studies. M.A. performed molecular dynamics simulations and molecular dynamics analysis. A.I.S. built molecular models. S.P.G., M.V.Y., K.D.N. and A.I.S. wrote the manuscript, which was then edited by all the authors. A.I.S. supervised the project.

**Competing interests** The authors declare no competing interests.

**Additional information**
**Correspondence and requests for materials** should be addressed to Alexander I. Sobolevsky.

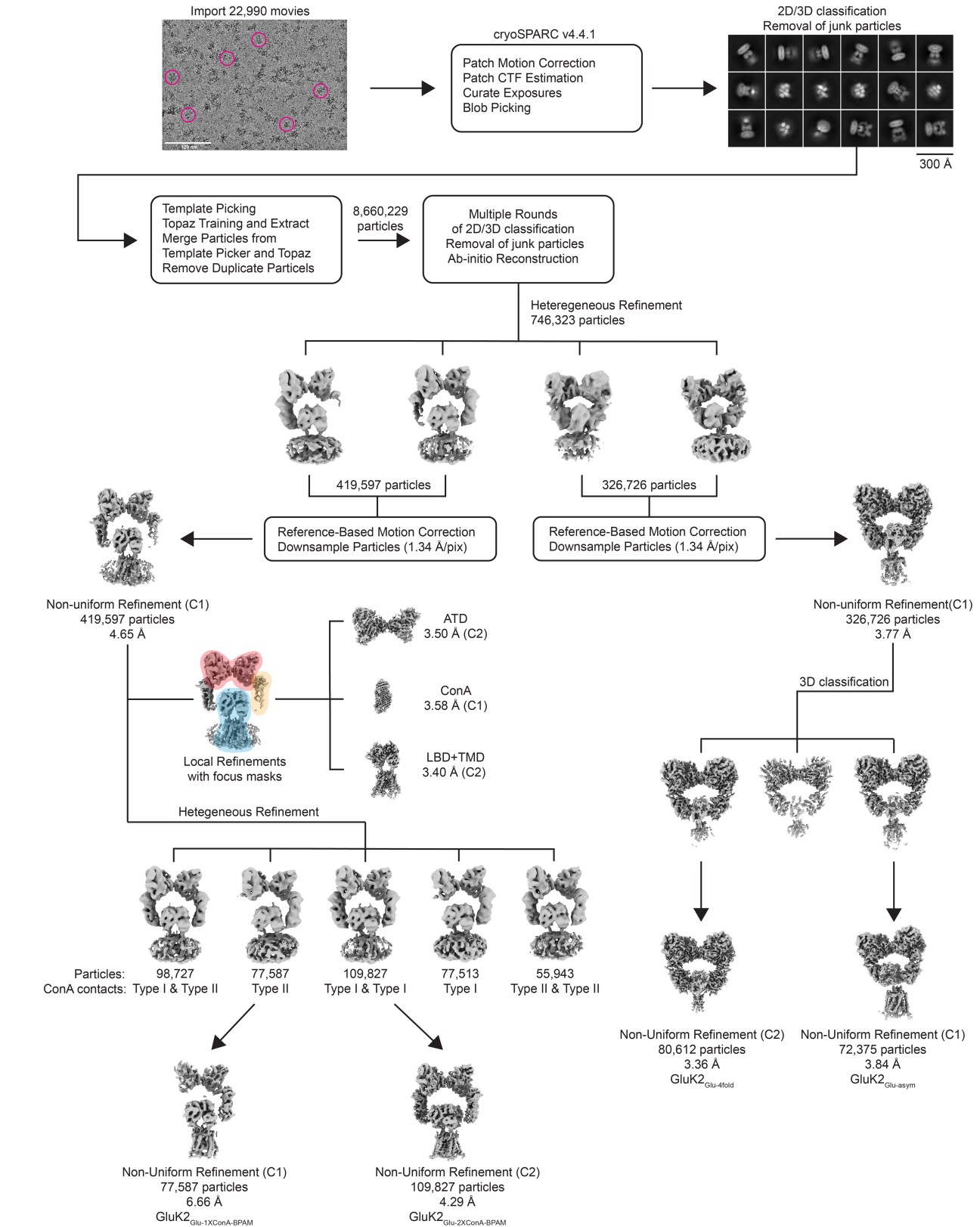

**Extended Data Fig.1 | Overview of the cryo-EM 3D reconstruction workflow.** On the top, a representative of the 22,990 collected micrographs with example particles circled in pink and 2D class averages.

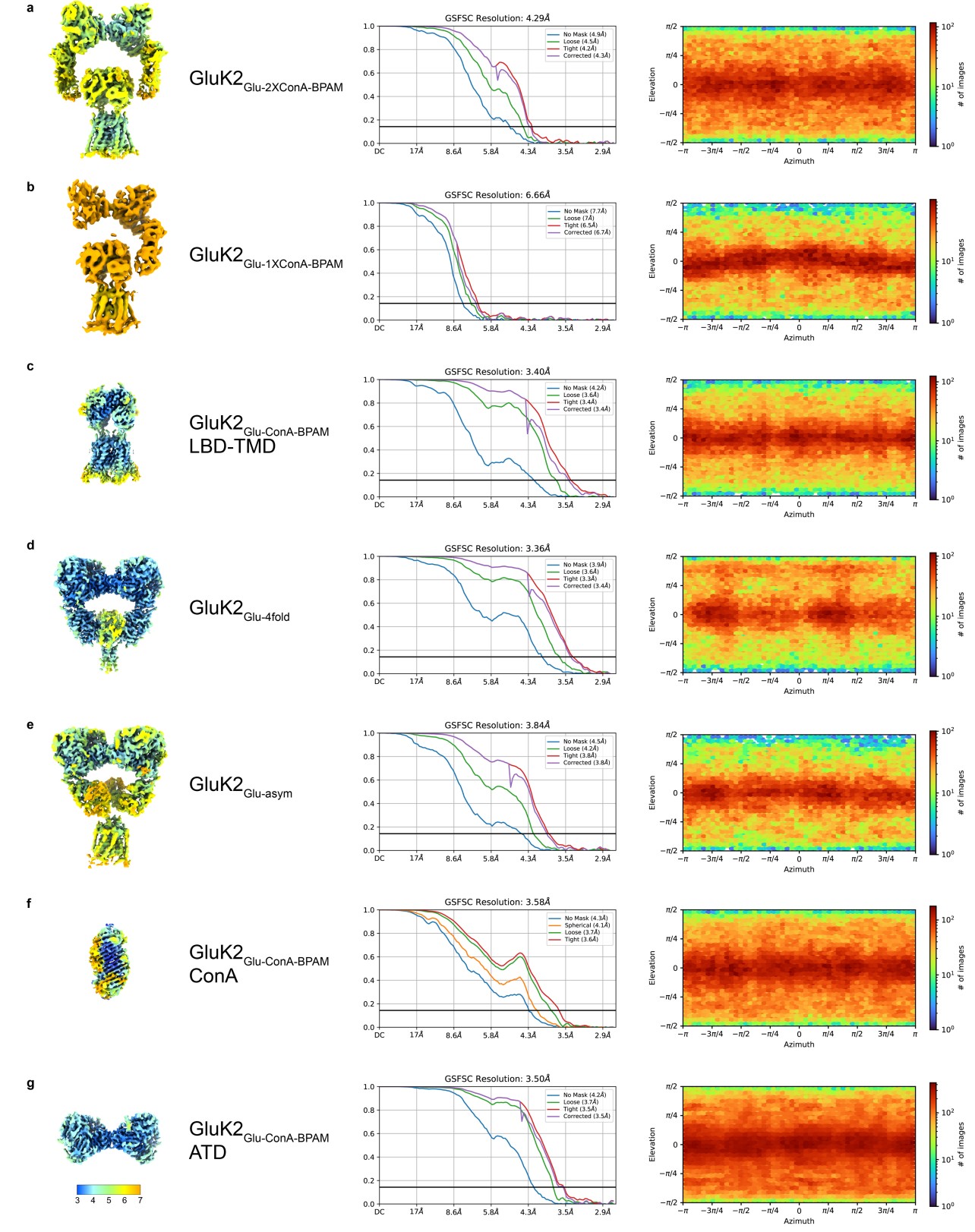

**Extended Data Fig. 2 | Characteristics of cryo-EM reconstructions.** Cryo-EM maps colored according to the local resolution estimation in cryoSPARC (left, color scale in Å), FSC curves calculated between half maps, with the overall resolution estimated using the FSC = 0.143 criterion (middle) and angular distribution of particles calculated using the 3D refinement reconstruction algorithm in cryoSPARC (right).

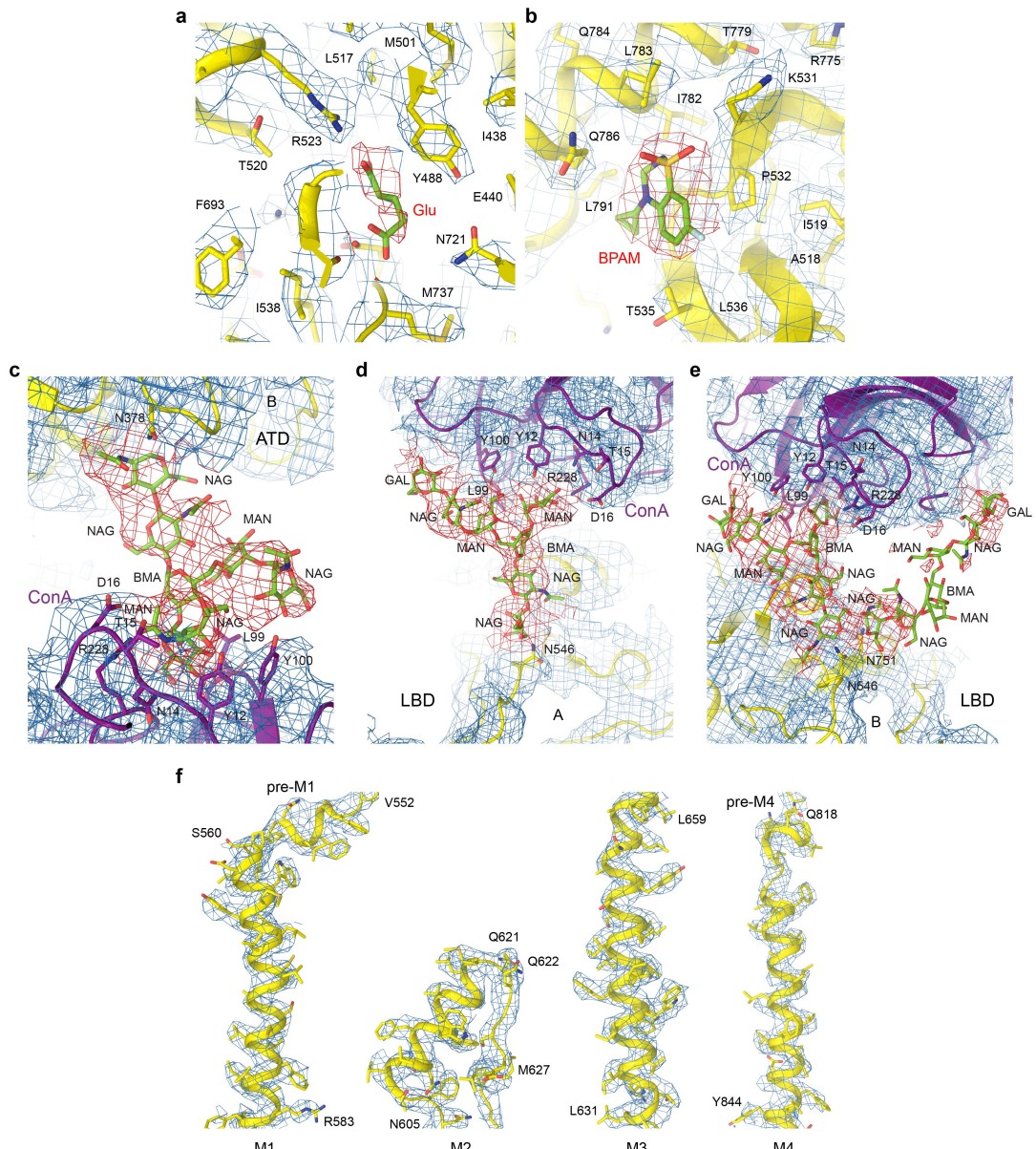

**Extended Data Fig. 3 | Cryo-EM density for GluK2_{Glu-ConA-BPAM}. a-b**, Binding pockets of Glu (**a**) and BPAM (**b**), with the protein structural model shown as a yellow ribbon and sticks, ligands as green sticks, and the corresponding density as blue and red mesh. **c-e**, ATD-ConA (**c**), ConA-LBD of type I (**d**), and ConA-LBD of type II (**e**) contact interfaces between the GluK2 and ConA, with the proteins shown as yellow and purple ribbons, respectively, their density as blue mesh, carbohydrates as green sticks and their density as red mesh. **f**, M1-M4 segments of the TMD.

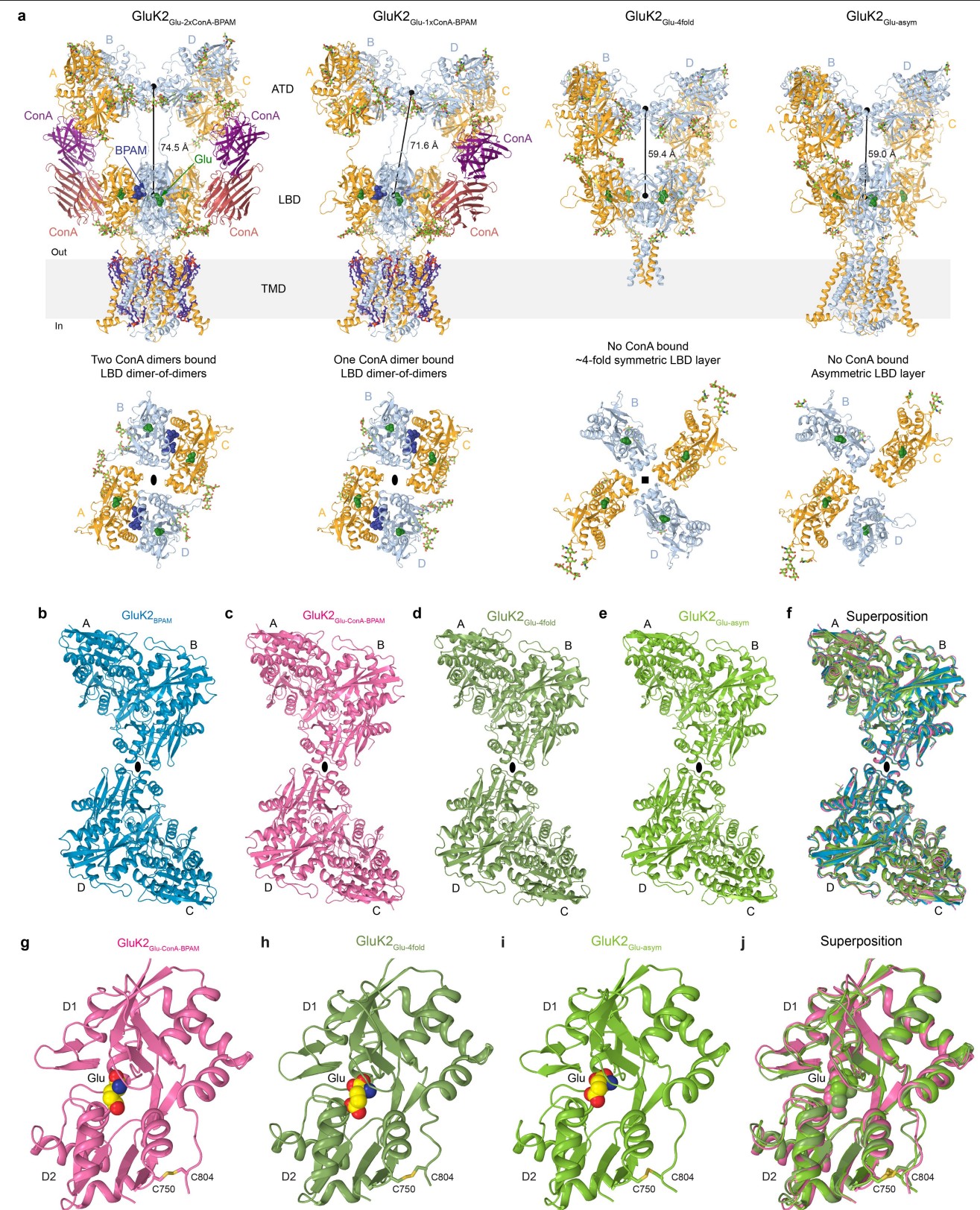

**Extended Data Fig. 4 | Structural ensemble observed for GluK2 in the presence of Glu, ConA and BPAM and domain comparisons. a**, Structures of GluK2$_{Glu-2xConA-BPAM}$, GluK2$_{Glu-1xConA-BPAM}$, GluK2$_{Glu-4fold}$ and GluK2$_{Glu-asym}$ viewed parallel to the membrane (top) and their LBD layers viewed extracellularly (bottom). Subunits A/C colored orange, B/D cyan, and ConA molecules purple and magenta. Molecules of Glu (dark green) and BPAM (dark blue) are shown as space-filling models. Carbohydrates (light green) and lipids (violet) are shown as sticks. The distances between the centers of mass of ATD and LBD layers are indicated. **b-f**, ATD layer structures from GluK2$_{BPAM}$ (**b**, PDB ID: 8FWS, blue), GluK2$_{Glu-ConA-BPAM}$ (**c**, pink), GluK2$_{Glu-4fold}$ (**d**, smudge), GluK2$_{Glu-asym}$ (**e**, green) and their superposition (**f**). **g-j**, Individual LBD structures from GluK2$_{Glu-ConA-BPAM}$ (**g**, pink), GluK2$_{Glu-4fold}$ (**h**, smudge), GluK2$_{Glu-asym}$ (**i**, green) and their superposition (**j**).

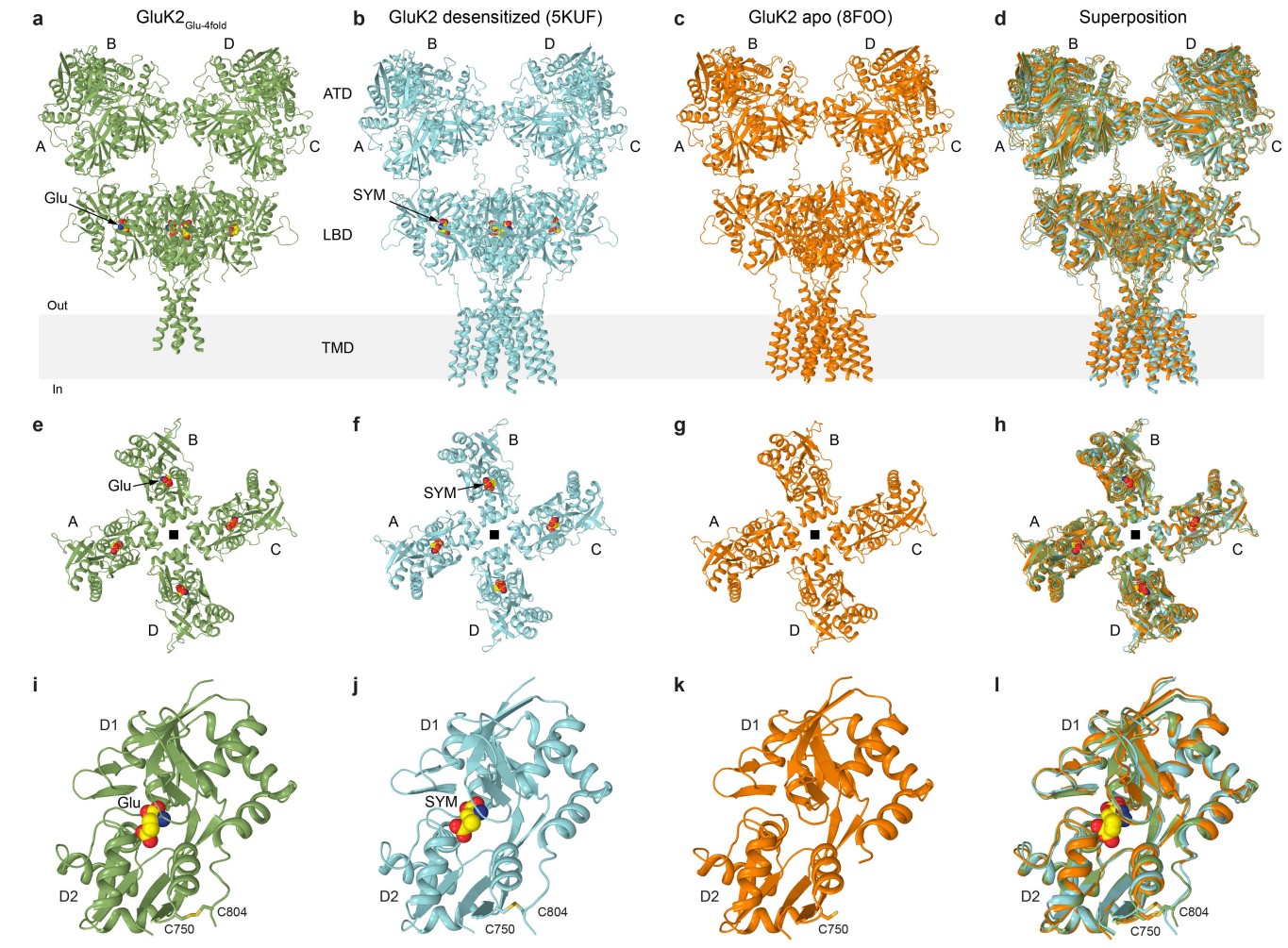

**Extended Data Fig. 5 | Comparison of GluK2 structures with 4-fold symmetrical LBD layer. a-d**, Structures of GluK2$_{Glu-4fold}$ (**a**, green), GluK2 in complex with (2 $S$,4 $R$)-4-methylglutamate (SYM) in the putative desensitized state (**b**, GluK2$_{SYM}$, PDB ID: 5KUF, cyan), GluK2 in the putative apo state (**c**, GluK2$_{apo}$, PDB ID: 8F0O, orange) and their superposition (**d**) viewed parallel to the membrane. The agonist molecules Glu and SYM are shown as space-filling models. **e-h**, LBD tetramers in GluK2$_{Glu-4fold}$ (**e**), GluK2$_{SYM}$ (**f**), GluK2$_{apo}$ (**g**) and their superposition (**h**) viewed extracellularly. **i-l**, Individual LBDs from GluK2$_{Glu-4fold}$ (**i**), GluK2$_{SYM}$ (**j**), GluK2$_{apo}$ (**k**) and their superposition (**l**).

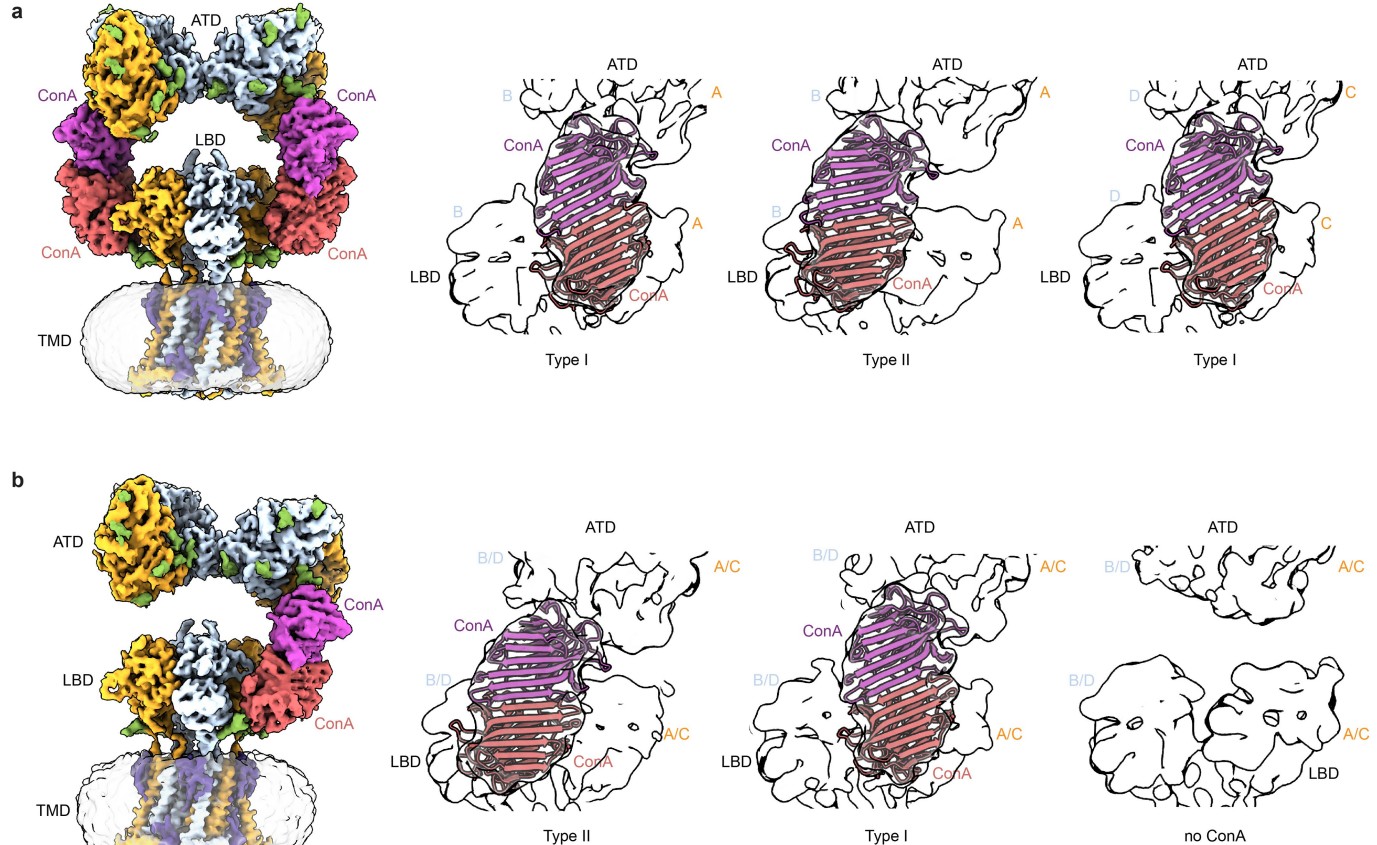

**Extended Data Fig. 6 | Different configurations of the ConA dimers connecting the ATD and LBD layers. a-b**, Cryo-EM density for GluK2$_{Glu-2xConA-BPAM}$ (**a**) and GluK2$_{Glu-1xConA-BPAM}$ (**b**) viewed from the side (left) and closeup views of different configurations of ConA dimers connecting ATD and LBD layers (right), with ConA molecules shown in ribbon representation. Note, there are two types of ConA-LBD contacts that are shown in more detail in Fig. 2f,g.

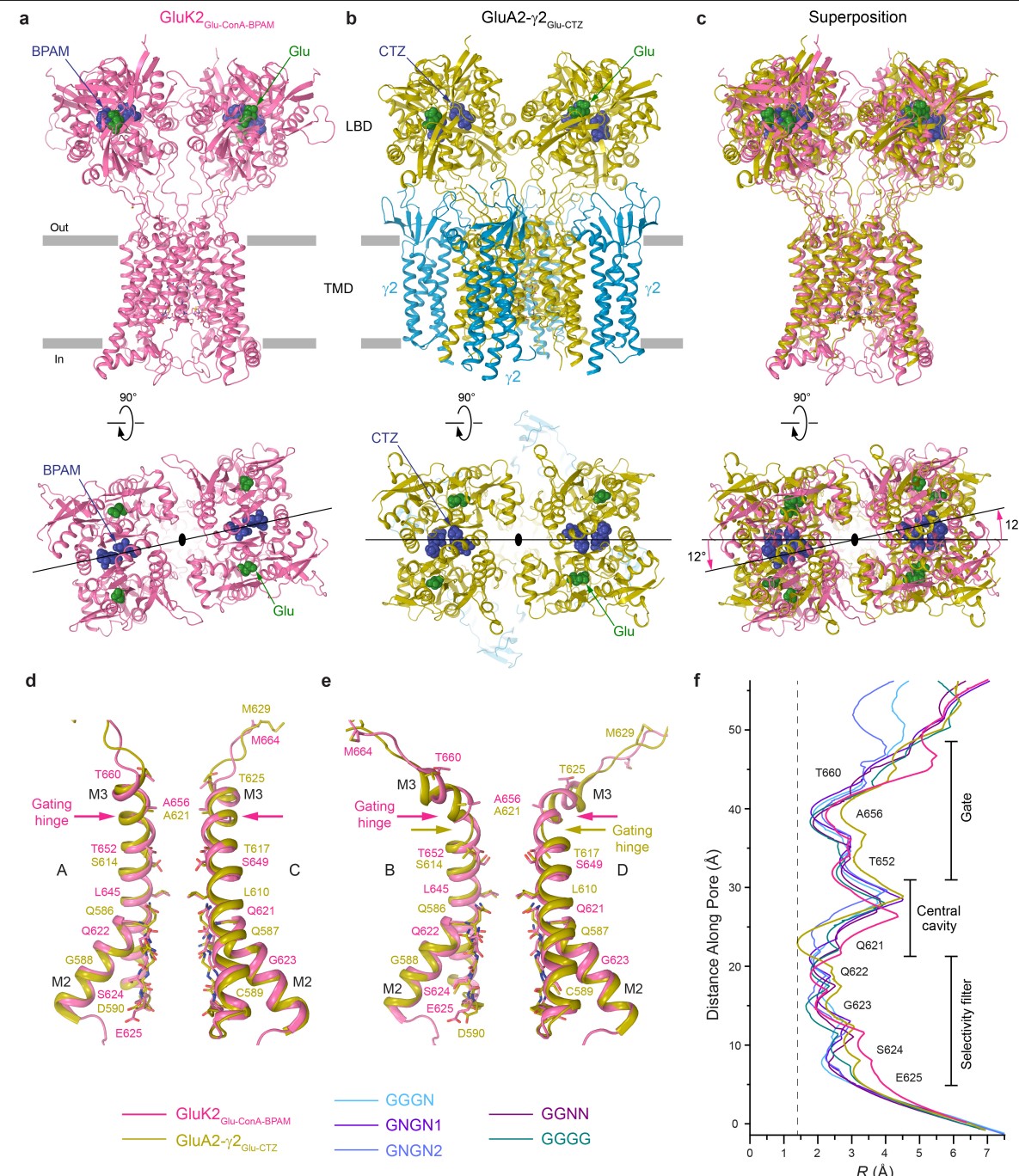

**Extended Data Fig. 7 | Comparison of open states in KARs and AMPARs.**
**a-c**, LBD-TMD structures in GluK2<sub>Glu-ConA-BPAM</sub> (**a**, pink), GluA2-γ2<sub>Glu-CTZ</sub> (**b**, PDB ID: 5WEO, olive and cyan) and their superposition (**c**) viewed parallel to the membrane (top) or extracellularly (bottom). The molecules of Glu and PAMs are shown as space-filling models. Pink arrows indicate a ~12° rotation of the LBD layer in GluK2<sub>Glu-ConA-BPAM</sub> relative to GluA2-γ2<sub>Glu-CTZ</sub>. **d-e**, Superposition of the pore-lining segments M2 and M3 in subunits A and C (**d**) and B and D (**e**) of

GluK2<sub>Glu-ConA-BPAM</sub> (pink), GluA2-γ2<sub>Glu-CTZ</sub> (olive), with residues lining the pore in the gate region shown as sticks. Arrows indicate the gating hinge in M3. **f**, Pore radius for GluK2<sub>Glu-ConA-BPAM</sub> (pink), GluA2-γ2<sub>Glu-CTZ</sub> (olive), GGGN (PDB ID: 7TNO, light blue), GNGN1 (PDB ID: 7TNL, purple), GNGN2 (PDB ID: 7TNM, blue), GGNN (PDB ID: 7TNN, violet) and GGGG (PDB ID: 7TNP, green) calculated using HOLE. The vertical dashed line denotes the radius of a water molecule, 1.4 Å.

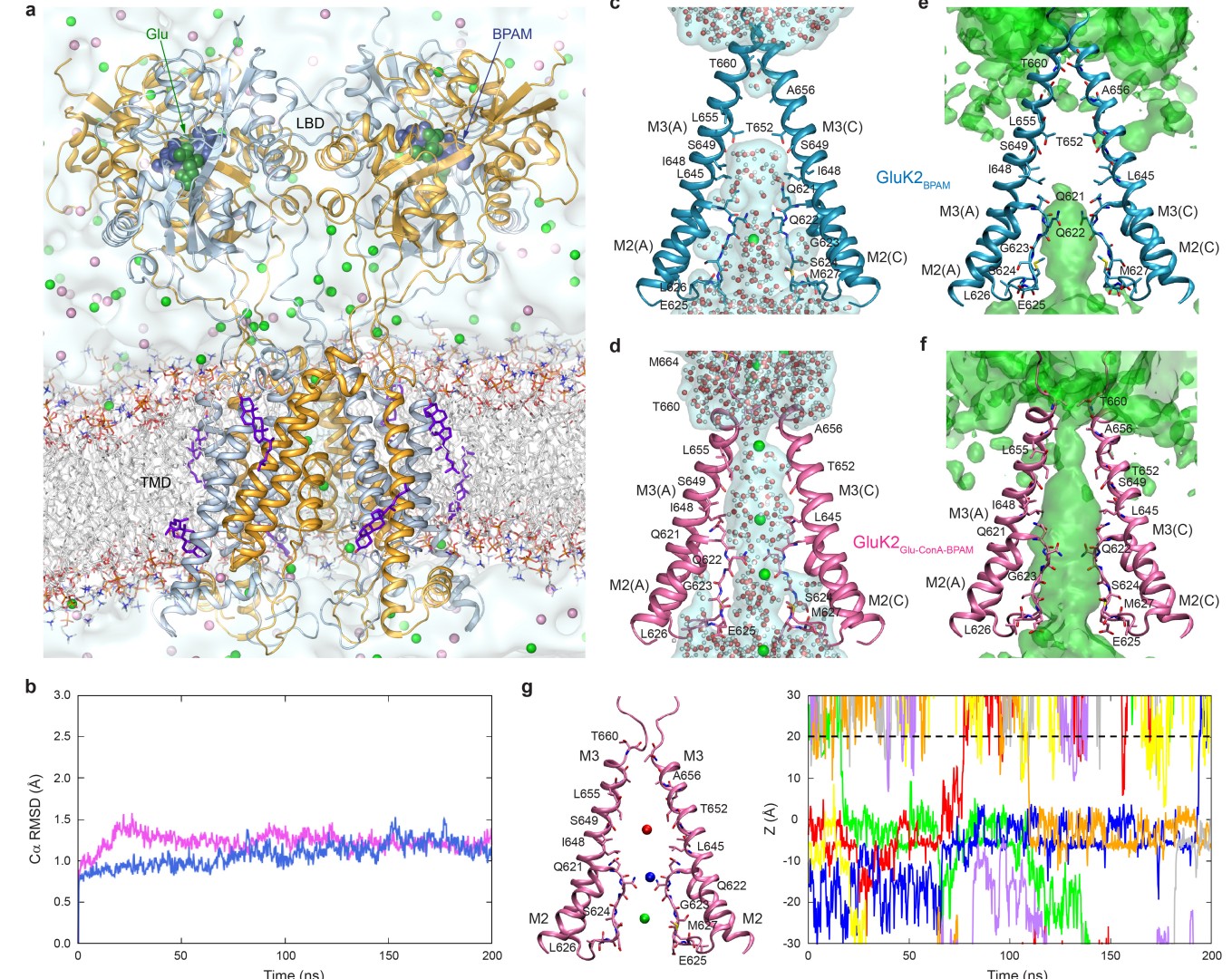

**Extended Data Fig. 8 | Molecular dynamics simulations. a**, A snapshot of the MD simulated GluK2_{Glu-ConA-BPAM} system in lipid bilayer and aqueous electrolyte solution, with the receptor subunits A/C shown as orange and B/D as cyan ribbons, lipid bilayer acyl chains in white and hydrophilic head groups as sticks, BPAM in dark blue and glutamate in dark green space-filling models, cholesterol as violet sticks, water as light blue continuum, and K+ and Cl- ions as green and pink spheres, respectively. **b**, Structural stability during MD simulations is illustrated as a time dependence of the Cα Root Mean Square Deviation (RMSD) for GluK2_{Glu-ConA-BPAM} (pink) and GluK2_{BPAM} (blue, PDB ID: 8FWS). **c-f**, Instant occupancy of water molecules (white/red spheres and light blue surface) and K+ ions (green spheres) (**c-d**) and cumulative density of K+ ions (green surface)

(**e-f**) for MD simulations of GluK2_{BPAM} (PDB ID: 8FWS) (**c,e**) and GluK2_{Glu-ConA-BPAM} (**d,f**). Pore-forming segments M2 and M3 (subunits A and C) in GluK2_{BPAM} (blue) and GluK2_{Glu-ConA-BPAM} (pink) are shown as ribbons, with residues lining the pore shown as sticks. **g**, On the left, the structure of GluK2_{Glu-ConA-BPAM} with three exemplar K+ ions colored differently. On the right, trajectories for K+ ions moving along the pore axis (Z) during the MD simulation of GluK2_{Glu-ConA-BPAM}. The same colors are used for the trajectories as for K+ ions represented by spheres on the left or in Supplementary Video 1. The dashed line indicates the gate region at z ~ 20 Å. K+ ions crossing this line permeate through the open gate. No K+ ion permeation was observed during MD simulations of GluK2_{BPAM}.

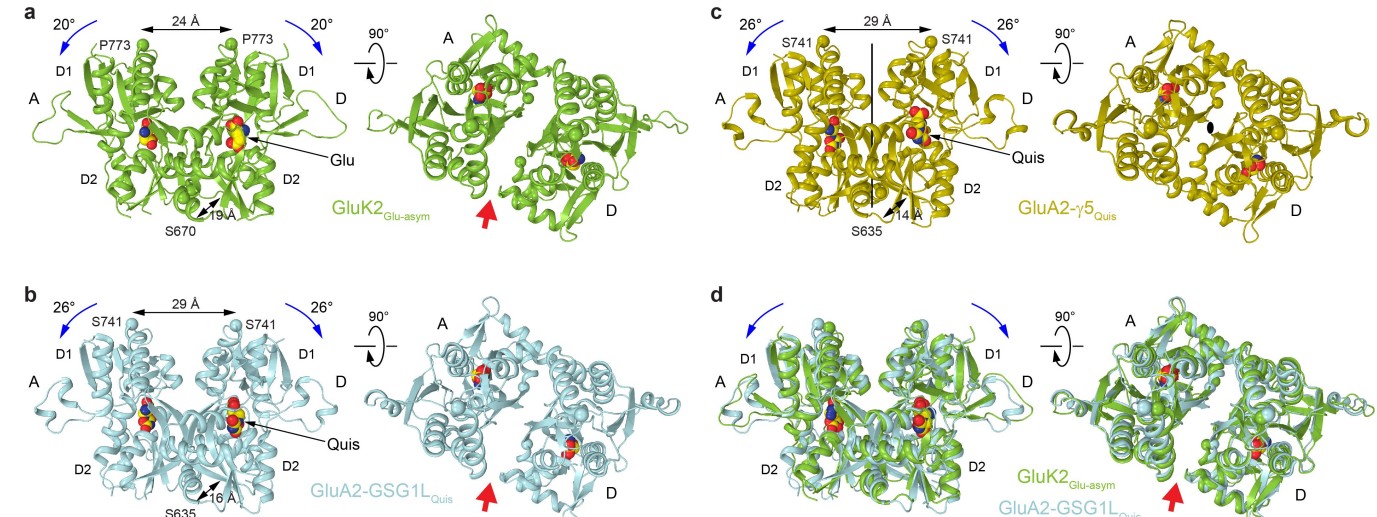

**Extended Data Fig. 9 | Comparison of desensitized-state dimers in KARs and AMPARs. a-d,** LBD dimers from structures of GluK2$_{Glu-asym}$ (**a**, green), GluA2-GSG1L$_{Quis}$ (**b**, PDB ID: 7RYZ, cyan), GluA2-γ5$_{Quis}$ (**c**, PDB ID: 7RZ8, olive) and superposition of GluK2$_{Glu-asym}$ and GluA2-GSG1L$_{Quis}$ (**e**) viewed from the side (left) or top (right). The agonist molecules Glu and quisqualate (Quis) are shown as space-filling models. Distances between Cα atoms of S670 and P773 in KARs and S635 and S741 in AMPARs are indicated. Rotation of the lobes D1 towards D2 upon agonist binding is illustrated by blue arrows. Red arrows point to the cleft between the LBD protomers signifying the loss of the dimer 2-fold rotational symmetry.

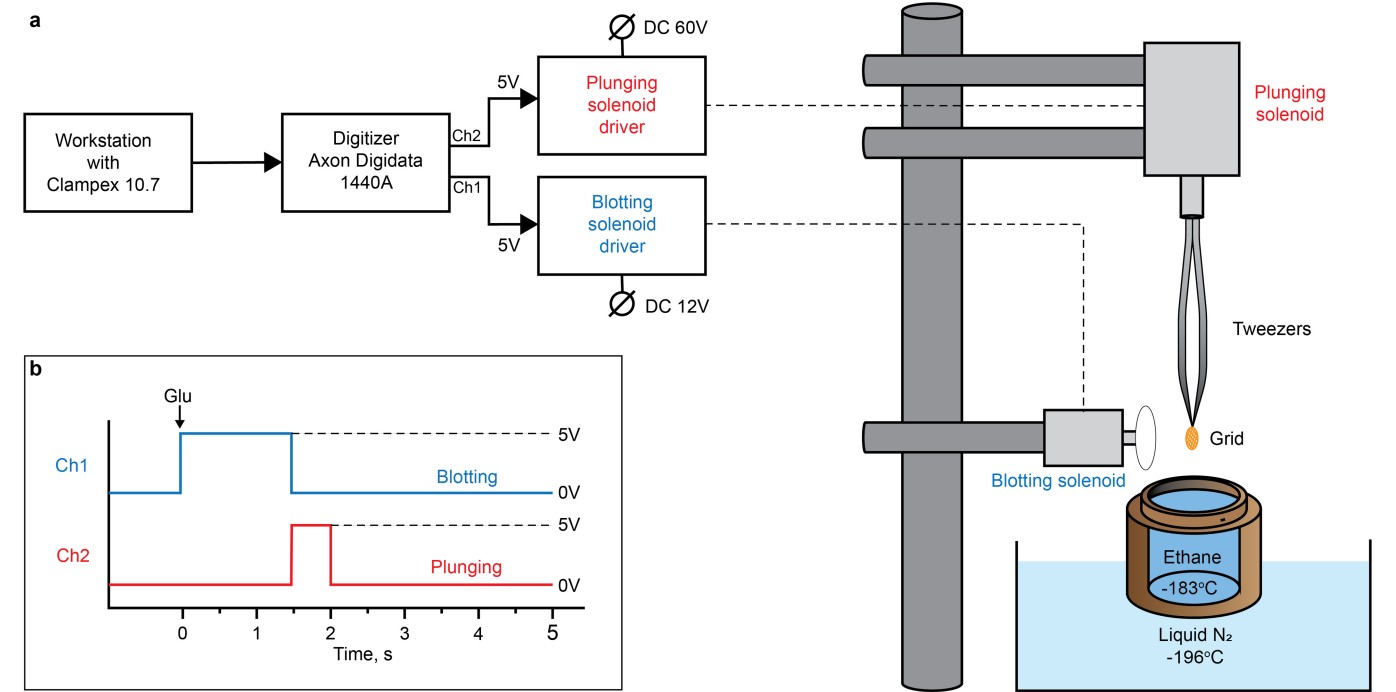

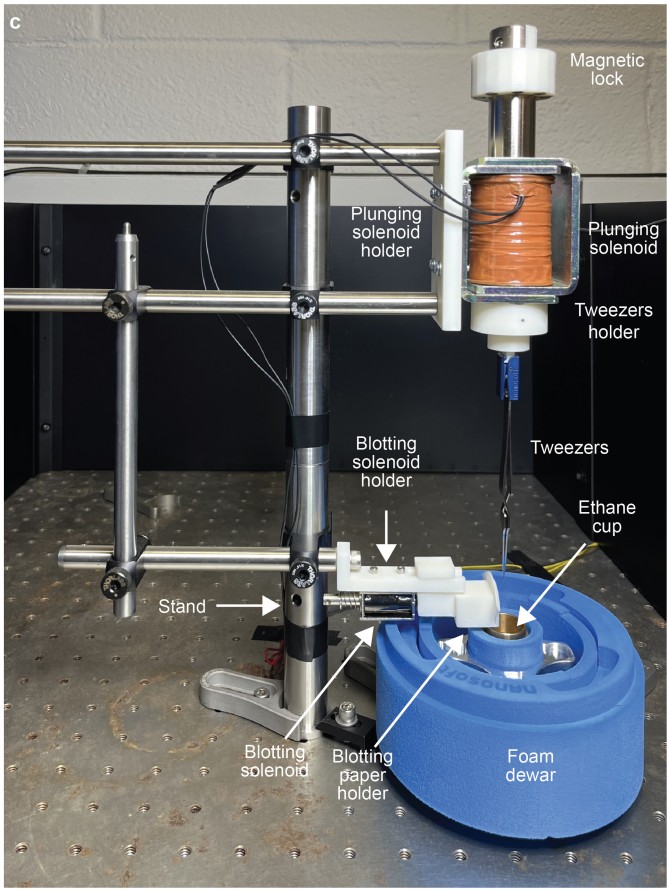

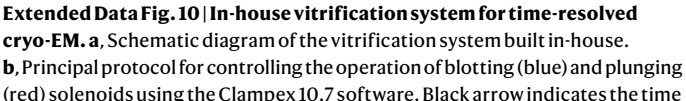

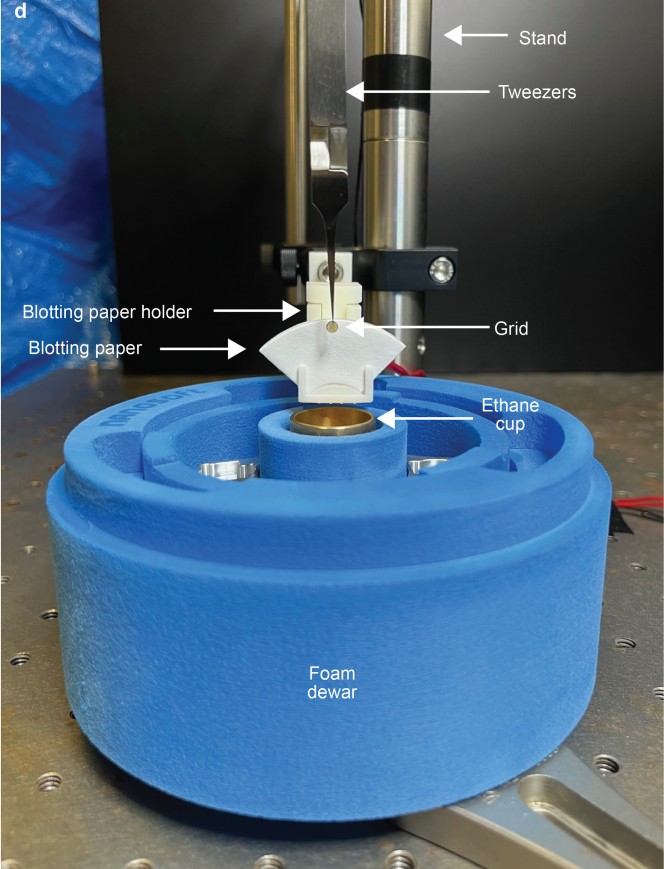

**Extended Data Fig. 10 | In-house vitrification system for time-resolved cryo-EM. a**, Schematic diagram of the vitrification system built in-house. **b**, Principal protocol for controlling the operation of blotting (blue) and plunging (red) solenoids using the Clampex 10.7 software. Black arrow indicates the time when glutamate was added. **c**, Overview of in-house vitrification system with the principal components labeled. **d**, Close-up view of blotting-freezing components of the system.

**Extended Data Table 1 | Cryo-EM data collection, refinement, and validation statistics**

| | GluK2 Glu-2xConA-BPAM | GluK2 Glu-1xConA-BPAM | GluK2 Glu-ConA-BPAM LBD-TMD | GluK2 Glu-4fold | GluK2 Glu-asym |
|---|---|---|---|---|---|
| EMDB code | EMD-44129 | EMD-44130 | EMD-44128 | EMD-44131 | EMD-44132 |
| PDB code | 9B36 | 9B37 | 9B35 | 9B38 | 9B39 |
| **Data collection and processing** | | | | | |
| Voltage (kV) | 300 | 300 | 300 | 300 | 300 |
| Electron exposure ($e^-Å^{-2}$) | 58 | 58 | 58 | 58 | 58 |
| Reported pixel size (Å/pix) | 1.34 | 1.34 | 1.34 | 1.34 | 1.34 |
| **Processing software** | cryoSPARC v4.4.1 | cryoSPARC v4.4.1 | cryoSPARC v4.4.1 | cryoSPARC v4.4.1 | cryoSPARC v4.4.1 |
| Symmetry imposed | C2 | C1 | C2 | C2 | C1 |
| Final particle images (no.) | 109,827 (EMD-44126) | 77,587 (EMD-44127) | 95,210 | 80,612 | 72,375 |
| Map resolution (Å) FSC 0.143 | 4.29 (EMD-44126) | 6.66 (EMD-44127) | 3.4 | 3.36 | 3.84 |
| **Refinement** | | | | | |
| Initial models used (PDB code) | 8FWQ, 3ENR | 8FWQ, 3ENR | 8FWS | 8FWQ | 8FWQ |
| FSC threshold | 0.143 | 0.143 | 0.143 | 0.143 | 0.143 |
| Map sharpening B factor ($Å^2$) | -171.8 (EMD-44126) | -557.4 (EMD-44127) | -91.4 | -88.4 | -82.3 |
| **Model composition** | | | | | |
| Non-hydrogen atoms | 36,342 | 32,654 | 15,472 | 22,670 | 27,522 |
| Protein residues | 4320 | 3846 | 1788 | 2720 | 3372 |
| **Ligands** | | | | | |
| 2J9 (BPAM344) | 4 | 4 | 4 | - | - |
| Glu (L-glutamate) | 4 | 4 | 4 | 4 | 4 |
| BMA ( D-mannose) | 18 | 13 | 4 | 12 | 8 |
| NAG (N-Acetylglucosamine) | 66 | 65 | 16 | 46 | 40 |
| POV (POPC) | 14 | 14 | 14 | - | - |
| CA (calcium) | 4 | 2 | - | - | - |
| GAL(D-galactose) | 4 | 4 | - | - | - |
| CLR (Cholesterol) | 8 | 8 | 8 | 16 | - |
| MAN (D-mannose) | 15 | 14 | - | - | 10 |
| ZN (zinc) | 4 | 2 | - | - | - |
| **B factors ($Å^2$)** | | | | | |
| Protein | 90.40 | 114.64 | 87.14 | 104.18 | 152.63 |
| Ligands | 61.35 | 75.56 | 35.93 | 42.98 | 91.72 |
| **R.m.s. deviations** | | | | | |
| Bond lengths (Å) | 0.007 | 0.007 | 0.008 | 0.007 | 0.009 |
| Bond angles (°) | 1.390 | 1.323 | 1.314 | 1.311 | 1.547 |
| **Validation** | | | | | |
| MolProbity score | 1.97 | 1.92 | 1.94 | 1.77 | 2.10 |
| Clash score, all atoms | 5.73 | 5.67 | 5.98 | 4.66 | 7.52 |
| Outliers rotamers (%) | 0.64 | 0.36 | 0.64 | 0.76 | 0.44 |
| **Ramachandran plot** | | | | | |
| Favored (%) | 85.21 | 87.55 | 87.39 | 90.59 | 83.18 |
| Allowed (%) | 14.05 | 11.98 | 11.71 | 9.04 | 15.36 |
| Outliers (%) | 0.74 | 0.47 | 0.90 | 0.37 | 1.46 |

# Reporting Summary

## Statistics

For all statistical analyses, confirm that the following items are present in the figure legend, table legend, main text, or Methods section.

| n/a | Confirmed | |
|---|---|---|
| ☐ | ☒ | The exact sample size (*n*) for each experimental group/condition, given as a discrete number and unit of measurement |
| ☐ | ☒ | A statement on whether measurements were taken from distinct samples or whether the same sample was measured repeatedly |
| ☐ | ☒ | The statistical test(s) used AND whether they are one- or two-sided *Only common tests should be described solely by name; describe more complex techniques in the Methods section.* |
| ☒ | ☐ | A description of all covariates tested |
| ☒ | ☐ | A description of any assumptions or corrections, such as tests of normality and adjustment for multiple comparisons |
| ☐ | ☒ | A full description of the statistical parameters including central tendency (e.g. means) or other basic estimates (e.g. regression coefficient) AND variation (e.g. standard deviation) or associated estimates of uncertainty (e.g. confidence intervals) |
| ☐ | ☒ | For null hypothesis testing, the test statistic (e.g. *F*, *t*, *r*) with confidence intervals, effect sizes, degrees of freedom and *P* value noted *Give P values as exact values whenever suitable.* |
| ☒ | ☐ | For Bayesian analysis, information on the choice of priors and Markov chain Monte Carlo settings |
| ☒ | ☐ | For hierarchical and complex designs, identification of the appropriate level for tests and full reporting of outcomes |
| ☒ | ☐ | Estimates of effect sizes (e.g. Cohen's *d*, Pearson's *r*), indicating how they were calculated |

*Our web collection on statistics for biologists contains articles on many of the points above.*

## Software and code

Policy information about availability of computer code

| Data collection | Leginon 3.5, pCLAMP 10.2 |
|---|---|
| Data analysis | cryoSPARC 4.4.1, UCSF Chimera 1.16, UCSF ChimeraX 1.3, COOT 0.9.8.1, PHENIX 1.18, pyMOL 2.5.2, HOLE 2.1, Origin 2023, pCLAMP 10.2, Clampfit 10.3, DynDom 1.5, CHARMM-GUI, AmberTools20, Amber20, VMD 1.9.4 |

For manuscripts utilizing custom algorithms or software that are central to the research but not yet described in published literature, software must be made available to editors and reviewers. We strongly encourage code deposition in a community repository (e.g. GitHub). See the Nature Portfolio guidelines for submitting code & software for further information.

## Data

Policy information about availability of data

All manuscripts must include a data availability statement. This statement should provide the following information, where applicable:
- Accession codes, unique identifiers, or web links for publicly available datasets
- A description of any restrictions on data availability
- For clinical datasets or third party data, please ensure that the statement adheres to our policy

The cryo-EM density maps have been deposited to the Electron Microscopy Data Bank (EMDB) under the accession codes EMD-44129 (GluK2Glu-2XConA-BPAM, composite map), EMD-44130 (GluK2Glu-1XConA-BPAM, composite map), EMD-44128 (GluK2Glu-ConA-BPAM, LBD-TMD), EMD-44131 (GluK2Glu-4fold), EMD-44132 (GluK2Glu-asym) , EMD-44125 (ConA, Type I), EMD-44124 (ConA, Type II), EMD-44123 (GluK2Glu-ConA-BPAM, ATD), EMD-44126 (GluK2Glu-2XConA-BPAM,

reference map), and EMD-44127 (GluK2Glu-1XConA-BPAM, reference map). The atomic coordinates have been deposited to the Protein Data Bank (PDB) under the accession codes 9B36 (GluK2Glu-2XConA-BPAM), 9B37 (GluK2Glu-1XConA-BPAM), 9B35 (GluK2Glu-ConA-BPAM, LBD-TMD), 9B38 (GluK2Glu-4fold), 9B39 (GluK2Glu-asym), 9B34 (ConA, Type I), and 9B33 (ConA, Type II). The atomic coordinates under the accession codes 3ENR, 5KUF, 5WEO, 7RZ8, 7RYZ, 7TNL, 7TNM, 7TNN, 7TNO, 7TNP, 8F0O, 8FWQ and 8FWS were used for model building and structural comparisons. Source data are provided with this paper.

# Research involving human participants, their data, or biological material

Policy information about studies with [human participants or human data](). See also policy information about [sex, gender (identity/presentation), and sexual orientation]() and [race, ethnicity and racism]().

| | |
|---|---|
| Reporting on sex and gender | N/A |
| Reporting on race, ethnicity, or other socially relevant groupings | N/A |
| Population characteristics | N/A |
| Recruitment | N/A |
| Ethics oversight | N/A |

Note that full information on the approval of the study protocol must also be provided in the manuscript.

# Field-specific reporting

Please select the one below that is the best fit for your research. If you are not sure, read the appropriate sections before making your selection.

☒ Life sciences  ☐ Behavioural & social sciences  ☐ Ecological, evolutionary & environmental sciences

For a reference copy of the document with all sections, see [nature.com/documents/nr-reporting-summary-flat.pdf](http://nature.com/documents/nr-reporting-summary-flat.pdf)

# Life sciences study design

All studies must disclose on these points even when the disclosure is negative.

| | |
|---|---|
| Sample size | Amount of cryo-EM data collected was limited by time allocation at the microscopes. For electrophysiological experiments, we only selected healthy-looking cells, with contrast body and smooth membrane that also showed the fluorescent signal of GFP expressed under a different from GluK2 promotor. No statistical approaches were used to predetermine the sample size but all measurements were performed using five or more biologically independent measurements. Exact number of biologically independent measurements and the number of independent experiments are reported in the figure legends. |
| Data exclusions | No data has been excluded. |
| Replication | No replication attempts have failed. Cryo-EM data collections were performed during two continuous two- and three-day data collection sessions and were consistent from the beginning to the end. A replication of the cryo-EM data collection was therefore not necessary or economically justifiable. In electrophysiological experiments, we made at least five independent replicates for each construct. |
| Randomization | Samples were not randomized; it is not technically or practically feasible to do so for cryo-EM or patch-clamp studies. Covariant control is not economically viable in cryo-EM data collections. Covariant control was also not possible in electrophysiological experiments due to the need to transfect with predetermind cDNAs and optimize protein expression for individual constructs. |
| Blinding | Researchers were not blinded; it is not technically or practically feasible to do so for cryo-EM or electrophysiological experiments. It is not economically viable to blind cryo-EM collections. For electrophysiological experiments, researchers conducting the studies were also in charge of cell as well as protein expression optimization for individual constructs in order to achieve recordings or transfected cells in these studies. These circumstances made blinding not possible. |

# Reporting for specific materials, systems and methods

We require information from authors about some types of materials, experimental systems and methods used in many studies. Here, indicate whether each material, system or method listed is relevant to your study. If you are not sure if a list item applies to your research, read the appropriate section before selecting a response.

## Materials & experimental systems

| n/a | Involved in the study |
|---|---|
| ☒ | Antibodies |
| ☐ | ☒ Eukaryotic cell lines |
| ☒ | Palaeontology and archaeology |
| ☒ | Animals and other organisms |
| ☒ | Clinical data |
| ☒ | Dual use research of concern |
| ☒ | Plants |

## Methods

| n/a | Involved in the study |
|---|---|
| ☒ | ChIP-seq |
| ☒ | Flow cytometry |
| ☒ | MRI-based neuroimaging |

# Eukaryotic cell lines

Policy information about cell lines and Sex and Gender in Research

| | |
|---|---|
| Cell line source(s) | HEK293S GnTI-, ATCC, Cat#CRL-3022<br>Sf9, Gibco, Cat#12659017<br>HEK 293, ATCC, Cat#CRL-1573 |
| Authentication | None of the cell lines used have been authenticated. |
| Mycoplasma contamination | The cell lines used have been tested for mycoplasma contamination by the providers (negative results) but have not been retested in the lab. |
| Commonly misidentified lines<br>(See ICLAC register) | No commonly misidentified lines were used in this study. |

# Plants

| | |
|---|---|
| Seed stocks | N/A |
| Novel plant genotypes | N/A |
| Authentication | N/A |

