## [Peer Review File · Nature]

Manuscript Title: Kainate receptor channel opening and gating mechanism

Reviewer Comments & Author Rebuttals

Reviewer Reports on the Initial Version:

Referees' comments:

Referee #1 (Remarks to the Author):

Kainate receptors (KARs), part of the iGluR family alongside AMPAR and NMDAR, play a crucial role in mediating excitatory neurotransmission and influencing neuronal circuits and synaptic plasticity. They are implicated in various neurological disorders. Despite their importance, KARs have received less attention compared to other iGluR family members. Previous studies, including those from this PI's lab and others, have identified several functional states of KARs. However, capturing the open state, crucial for understanding channel gating mechanisms, has remained elusive, primarily due to the challenges posed by KAR's rapid desensitization and the limitations of cryo-electron microscopy (cryo-EM) in capturing this extremely transient state.

This study by Gangwar et al. took advantage of the combined effect of positive modulators ConA and BPAM to slow down the desensitization rate of GluK2 to seconds level. They further took an innovative application of time-resolved cryo-EM to freeze the sample within 2-3s after applying agonist Glutamate. These together allowed them to capture various functional states of GluK2, including the transient open state. This manuscript is well written and unveils groundbreaking insights into the gating mechanism of GluK2 and iGluR. Despite similar mechanism underlying channel opening to the AMPAR receptor, the authors captured the open state with maximum ion conductance (O4) among the iGluR family with kinking of all four M3 helices, providing a comprehensive and significant advancement in our understanding of channel gating in iGluRs. Moreover, this work is a valuable contribution not only to the field of iGluR, but also has implications that extend to the ion channel field and broader scientific community who are interested in time-resolved cryo-EM techniques to capture transient states.

While the study is solid and comprehensive, I do have a few questions that need to be addressed. A few areas could benefit from additional clarification to further enhance this already excellent manuscript.

1. The role of glycosylation sites in ConA binding is interesting. Could the authors provide experimental results whether mutations at these sites affect GluK2 expression level? Additionally, it would be informative to know if these residues are conserved across KAR subtypes and if ConA exhibits specificity towards GluK2 type. Clarification on how these glycosylation sites are defined and their potential roles in the absence of ConA would also be beneficial.
2. ConA and BPAM together markedly slow down the KAR desensitization, however, the individual roles of ConA and BPAM and their cooperativity in KAR activation remain ambiguous from the structural analysis in the manuscript. A more detailed explanation and discussion would be helpful.

3. The manuscript described two structures of GluK2 without ConA binding in symmetric and asymmetric forms. Despite the addition of BPAM, no BPAM was observed, obviously caused by the abruption of the BPAM site in LBD. However, the previous findings from the same lab showed the BPAM molecule in the GluK2 structure. Could the authors elaborate on the distinctions between these structures and discuss the potential reasons of missing a class with BPAM only?
4. The transmembrane domains and lipids in the two GluK2 structures without ConA are either missing or not clearly defined. However, the previously published GluK2 structures have well-defined TMD and lipids. Could the authors provide an explanation for the disordered transmembrane domain in this state?
5. The ATD-LBD linker exhibits flexibility to accommodate regulatory molecules like ConA. How the presence or absence of ConA binding affects the linker. Specifically, how does ConA binding alter the structure of dynamics of the ATD-LBD linker, and what implication does this have for channel modulation?
6. Two types of ConA-LBD interaction are observed, type I and type II (Fig. 2f and 2g). However, it is unclear how these two types are defined in the structure as the figure 2a only shows interaction in Figure 2f. Are they from two different classes? How the interacting residues such as L99, Y100 or R228 in ConA affect the modulation effect on GluK2?
7. What are the differences between 1x-ConA and 2x-ConA structure? It is not evident from the current description if both configurations lead to open states or if they differ in their pore profiles (Fig 4). Additionally, in the GluK2Glu-1xConA-BPAM structure, it would be beneficial to understand how ConA's presence influences BPAM binding. Does the loss of one ConA molecule alter the LBD configuration compared to the GluK2Glu-2xConA-BPAM structure? Is one ConA sufficient to open the channel?

Minor:

1. It is unclear about the functional state of the asymmetrical GluK2Glu structure, does it undergo a channel desensitization from open state induced by Glu, ConA and BPAM?
2. Keep the nomenclature of BPAM in consistency throughout the text. E.g. Line 103: "Given the synergistic effect of PAMs on the current amplitude".
3. Extended Data Fig. 3. Densities of glycosylation sites and carbohydrates linked to ConA binding should be shown.

Juan Du

Referee #2 (Remarks to the Author):

This is a neat paper with great figures that expands our view of glutamate receptor activation. Using an elegant time-resolved approach calibrated to functional measurements, the authors have managed to catch the GluK2 kainate receptor in a variety of activation states, including, against the odds, an open state. Technical challenges are, as usual with this expert group, met and overcome.

Major points

1. What is new here? We get a new open arrangement, the first in kainate receptors, and it's a bit different from AMPA receptors. The authors conclude that this open kainate receptor channel must conduct ions and water because AMPA receptors that are open do so in MD simulations. I thought it's rather a shame not to show this simple result computationally (if indeed that is what happens). Not all activated AMPA channels conduct in MD simulations, even those that look as if they really should, so this is not really an open and shut case. Collapsing the pore dimensions to a one dimensional profile is often done but doesn't really answer this point. The current presentation of this data doesn't do a good job of bringing out the novelty.

2. The authors do update our understanding of the lectin Concanavalin A considerably, but do not reference previous work that would indicate the extent of the advance. In somewhat implausible but influential work, it was shown that one glycan, any glycan -even an ectopic site -, was all that was needed for the activity of ConA (Everts I, Petroski R, Kizelsztejn P, Teichberg VI, Heinemann SF, Hollmann M (1999) Lectin-induced inhibition of desensitization of the kainate receptor GluR6 depends on the activation state and can be mediated by a single native or ectopic N-linked carbohydrate side chain. *J Neurosci* 19(3):916–927.)^[1]_{SEP} This study was odd because it suggested instead of a clear structural mechanism (as demonstrated here) rather something non-specific breaking the receptor desensitization, perhaps because ConA is a “dirty” ligand. It would be good to cite this work (and perhaps later work from Derek Bowie) that would indicate the confusion and difficulty in this area. The work in hand does a lot to clear up this confusing picture, and for the first time explains how ConA works.

3. There are plenty of inactive state structures already known for other glutamate receptors, so the menagerie of extra structures provided here is merely welcome, rather than revolutionary.^[1]_{SEP}

4. The authors write that “Our structures reveal the molecular basis of KAR gating that determines the unique role of these receptors in neurobiology and disease” which is hyperbole built on exaggeration. I didn't find anything in this paper to help in this respect. We could imagine from the new structures how to target kainate receptors with novel therapeutics, which the authors mention. The provided list of neurological disorders that kainate receptors have been linked to is long, and the authors reference some reviews on the topic. I think there are as many reviews as original research works in this area. And most of the reviews feature a lot of synaptic physiology and then some “maybe” sections at the end about pathological states. Most of the genetics studies have not been followed up. I know that it would be nice to think that these structures explain kainate receptors in the brain, but they don't add much to an already very scarce pool of information. A particular concern in this regard is that almost no receptors in the brain are GluK2 homomers, so the connection to physiology or pathology is even more tenuous. Heteromer structures of kainate receptors were already published.

5. The authors cite two open access designs for plungers, one that is light coupled. Perhaps the authors could give a bit more detail? The methods section does not read naturally and is, to my eye, inadequate.

“The operational principles of the in-house vitrification system align with a time-resolved vitrification device(45,46) previously outlined with minor modifications.”

One could also argue that the time-resolved aspect is somewhat overplayed, coming in at 3 seconds. But, the functional measurements show that this is an appropriate timescale. In any case, the technological aspect here is the most important, and should be described more completely.

Minor points

* The authors do not mention the RNA editing status of the construct.

* 3 references (21-23) are cited in a block of 11 at the end of the following sentence

“Both principal and auxiliary subunits determine KAR functional properties, including ion permeation, channel block, ligand specificity and kinetics, and play important roles in shaping KAR-mediated synaptic responses”

These three references, all from Derek Bowie’s lab, are about ion modulation and don’t seem to bear on the properties discussed. They were largely superseded by structures of the ion binding sites. But maybe the authors find particular value in these works that I missed.

Author Rebuttals to Initial Comments:

We thank Reviewers for their excellent suggestions that have led to a significant improvement of this manuscript. We have made changes in the manuscript with details outlined in our responses below.

Referee #1 (Remarks to the Author):

Kainate receptors (KARs), part of the iGluR family alongside AMPAR and NMDAR, play a crucial role in mediating excitatory neurotransmission and influencing neuronal circuits and synaptic plasticity. They are implicated in various neurological disorders. Despite their importance, KARs have received less attention compared to other iGluR family members. Previous studies, including those from this PI's lab and others, have identified several functional states of KARs. However, capturing the open state, crucial for understanding channel gating mechanisms, has remained elusive, primarily due to the challenges posed by KAR's rapid desensitization and the limitations of cryo-electron microscopy (cryo-EM) in capturing this extremely transient state.

This study by Gangwar et al. took advantage of the combined effect of positive modulators ConA and BPAM to slow down the desensitization rate of GluK2 to seconds level. They further took an innovative application of time-resolved cryo-EM to freeze the sample within 2-3s after applying agonist Glutamate. These together allowed them to capture various functional states of GluK2, including the transient open state. This manuscript is well written and unveils groundbreaking insights into the gating mechanism of GluK2 and iGluR. Despite similar mechanism underlying channel opening to the AMPAR receptor, the authors captured the open state with maximum ion conductance (O4) among the iGluR family with kinking of all four M3 helices, providing a comprehensive and significant advancement in our understanding of channel gating in iGluRs. Moreover, this work is a valuable contribution not only to the field of iGluR, but also has implications that extend to the ion channel field and broader scientific community who are interested in time-resolved cryo-EM techniques to capture transient states.

We thank Reviewer #1 for the generous assessment of our work.

While the study is solid and comprehensive, I do have a few questions that need to be addressed. A few areas could benefit from additional clarification to further enhance this already excellent manuscript.

1. The role of glycosylation sites in ConA binding is interesting. Could the authors provide experimental results whether mutations at these sites affect GluK2 expression level? Additionally, it would be informative to know if these residues are conserved across KAR subtypes and if ConA exhibits specificity towards GluK2 type. Clarification on how these glycosylation sites are defined and their potential roles in the absence of ConA would also be beneficial.

To assess the expression level of the glycosylation site mutants, we have plotted the maximal current amplitude measured in whole-cell patch-clamp experiments.

Neither one of the mutants showed statistically significant difference in the maximal current amplitude compared to wild type (two-sided two-sample *t*-test, $P > 0.1$) suggesting that the mutations did not significantly affect the expression level. We have also made an alignment of the GluK subunits:

Yellow are glycosylated ASN residues and **cyan** are N-X-T/S residues

```

sp | P22756 | GRIK1_RAT      MERSTVLIQPLWTRDTSWTLTYFLCY---ILPQTSPQVLRIGGIFETVENE---PVNVE 54
sp | P42260 | GRIK2_RAT      MKIISPVLNLFVFSRS----IKVLLCLLWIGYSQGTTHVLRFGGIFEFVESG---PMGAE 53
sp | P42264 | GRIK3_RAT      MTAPWRRRLRSLVWEYW---AGFLVCAFWIPDSRGMPHVIRIGGIFEFYADGPNAQVMNAE 56
sp | Q01812 | GRIK4_RAT      -----MPRVSAPLVLLPAWLLMVACSPHSLRIAAILDDPME----CSRGE 41
sp | Q63273 | GRIK5_RAT      -----MPAEL-LLLLIVAFANPSCQVLSLRLMAAILDDQTV----CGRGE 40
                                     .:                               *:..*::          *

sp | P22756 | GRIK1_RAT      ELAFKFAVTSINRNTLMPNTTLTYDIQRINLFDSEASRRACDQLALGVAALFGPSHSS 114
sp | P42260 | GRIK2_RAT      ELAFRFVAVNTINRNTLLPNNTTLTYDTQKINLYDSFEASKKACDQLSLGVAAIFGPSHSS 113
sp | P42264 | GRIK3_RAT      EHAFRFSANIINRNTLLPNNTTLTYDIQRIHFHDSFEATKKACDQLALGVVAIFGPSQGS 116
sp | Q01812 | GRIK4_RAT      RLSITLAKNRINRAPERLGKAKVEVDIFELLRDSEYETAETMCQILPKGVAVLGPSSSP 101
sp | Q63273 | GRIK5_RAT      RLALALAREQINGIIEVPAKARVEVDIFELQRDSQYETTTDTCQILPKGVSVLGPSSSP 100
. : : : **      : : : *  . :  . : * : : * * . : : * * * .

sp | P22756 | GRIK1_RAT      -SVSAVQSICNALEVPHIQTRWKHPS-VDSRDLFYINLYPDYAAISR AVL DLVLYNWK T 172
sp | P42260 | GRIK2_RAT      -SANAVQSICNALGVPHIQTRWKHQV-SDNKDSFYVSLYPDFSSLSRAILDVQFFKWK T 171
sp | P42264 | GRIK3_RAT      -CTNAVQSICNALEVPHIQLRWKHHP-LDNKDTFYVNLYPDYASLSHAILDLVQSLKWR S 174
sp | Q01812 | GRIK4_RAT      ASSSIISNICGEKEVPHFKVAPEEFVRFQLQRFTTLNLHPSNTDISVAVAGILNFNCTT 161
sp | Q63273 | GRIK5_RAT      ASASTVSHICGEKEIPHIVKGPETPRQLYLRFASVSLYPSNEDVSLAVSRILKSFNYPS 160
. . . : ** .  : * : : . :  :  . : * : .  : * * : : : :

sp | P22756 | GRIK1_RAT      VTVVYEDSTGLIRLQELIKAPSRYNIKIKIRQLPPANKDAKPLLKEMKKSKEFYVIFDCS 232

```

sp | P42260 | GRIK2_RAT VTVVYDDSTGLIRLQELIKAPSRYNLRKIRQLPADTKDAKPLLKEMKRGKEFHVIFDCS 231
 sp | P42264 | GRIK3_RAT ATVVYDDSTGLIRLQELIMAPSRYNIRLKIQLPIDSDDSRPLLKEMKRGREFRIIFDCS 234
 sp | Q01812 | GRIK4_RAT ACLICAKAECLLNLEKLLRQFLISKDTLSVRM-LDDTRDPTPLLKEIRDDKTATIIIHAN 220
 sp | Q63273 | GRIK5_RAT ASLICAKAECLLRLEELVRGFLISKETLSVRM-LDDSRDPTPLLKEIRDDKVSTIIIDAN 219
 . : : . : * : * : * : * : : : : * * * * * : : : : * : . . .

sp | P22756 | GRIK1_RAT HETAAEILKQILFMGMMTEYYHYFTTLDLFALDLELYRSGV NMT GFRKLNIDNPHVSS 292
 sp | P42260 | GRIK2_RAT HEMAAGILKQALAMGMMTEYYHYIFTTLDLFALDVEPYRSGV NMT GFRILNNTENTQVSS 291
 sp | P42264 | GRIK3_RAT HTMAAQILKQAMAMGMMTEYYHYIFTTLDLYALDLEPYRSGV NLT GFRILNVDNAHVSA 294
 sp | Q01812 | GRIK4_RAT ASMSHTILLKAAELGMVSAYTYIFTNLEFSLQRMDSLVDDRNVNIGFSIF NQS HAFFQE 280
 sp | Q63273 | GRIK5_RAT ASISHLVLRKASELGMTSAFYKYILTMDFPILHLDGIVEDSSNILGFSMF NTS HFPYPE 279
 : : * : : * * : : * : * : * : * : : : . * : * * : * . :

sp | P22756 | GRIK1_RAT IIEKWSME---RLQAPRPETGLLDGMMTTEAALMYDAVYMVAIASH---RASQLTVSSL 346
 sp | P42260 | GRIK2_RAT IIEKWSME---RLQAPPKPDGSLLDGFMTDAALMYDAVHVSVAVQ---QFPQMTVSSL 345
 sp | P42264 | GRIK3_RAT IVEKWSME---RLQAAPRAESGLLDGVMMDAALLYDAVHIVSVCYQ---RASQMTVNSL 348
 sp | Q01812 | GRIK4_RAT FSQSL NQSWQENCDHVPFTGP-----ALSSALLFDAVYAVVTAVQEL NRS QEIGVKPL 333
 sp | Q63273 | GRIK5_RAT FVRSLN MSWRENCEASTYGP-----ALSAALMFDAVHVVS AVREL NRS QEIGVKPL 332
 : : : * * * * * : * . : : : * . *

Asn378 (GluK2)

sp | P22756 | GRIK1_RAT QCHRHKPCALGPRFMNLIKEARWDGLTGRITFNKT DGLRDKDFLDL IISLKEEGTEKASGE 406
 sp | P42260 | GRIK2_RAT QCNRHKPWRFGTRFMSLIKEAHWEGLTGRITFNKT NGLRTDFDLDV IISLKEEGLE----- 400
 sp | P42264 | GRIK3_RAT QCHRHKPWRFGGRFMNFIKEAQWEGLTGRIVFNKT SGLRTDFDLDI IISLKEDGLE----- 403
 sp | Q01812 | GRIK4_RAT SCGSAQIQHGTSLMNYLRMVELEGLTGHI EFNS-KGQRSNYALKILQFTRNGFR----- 387
 sp | Q63273 | GRIK5_RAT ACTSANIWPHGTSMLMNYLRMVEYDGLTGRV EFNS-KGQR TNYTLRILEKSRQGHR----- 386
 * : * : * . : . : * * * * * : * . * * . : * : . . . : * .

sp | P22756 | GRIK1_RAT VSKHLYKVWKIGIWNSSNGL NMT DGNRDRSNNIT DSLAN R TLIVTTILEEPYVMYRKSD 466
 sp | P42260 | GRIK2_RAT -----KIGTWDPASGLN MTE SQKGPANIT D SLS NRS LIVTTILEEPYVLFKSD 450
 sp | P42264 | GRIK3_RAT -----KVGWVSPADGLNITEVAKGRGPNVIT D SLT NRS LIVTTILEEPVFMFRKSD 453
 sp | Q01812 | GRIK4_RAT -----QIGQWHVAEGLSMDSR--LYASNIS D S LFN TLVVTTILENPYMLKGNH 435
 sp | Q63273 | GRIK5_RAT -----EIGVWYSNRTLAM NAT --TLDINLS QTLANKTLVVTILENPYVMRRPNF 434
 : * * * : * : * * * * * : * * * * * : * * * * * : .

sp | P22756 | GRIK1_RAT KPLYGNDRFEAYCLDLLKELSNILGFLYDVKLVDPDGKYGAQND-KGEWNGMVKELIDHRA 525
sp | P42260 | GRIK2_RAT KPLYGNDRFEGYCIDLLRELSTILGFTYEIRLVEDGKYGAQDDVNGQWNGMVRELIDHKA 510
sp | P42264 | GRIK3_RAT RTLYGNDRFEGYCIDLLKELAHILGFSYEIRLVEDGKYGAQDD-KGQWNGMVKELIDHKA 512
sp | Q01812 | GRIK4_RAT QDMEGNDRYEGFCVDMLKELAEILRFNYKIRLVGDGVYGVPEA-NGTWTGMVGEIARKA 494
sp | Q63273 | GRIK5_RAT QALSGNERFEGFCVDMLRELAEELLRFYRLRLVEDGLYGAPEP-NGSWTGMVGEINRKA 493
: : **:*:*_*:*:*:*:*: * * * : ** * * * . : * * . ** * * : **

Asn546 (GluK2)

sp | P22756 | GRIK1_RAT DLAVAPLTITYVREKVIDFSKPFMTLGISILYRKPNGTNPVFSFLNPLSPDIWMYVLLA 585
sp | P42260 | GRIK2_RAT DLAVAPLAITYVREKVIDFSKPFMTLGISILYRKPNGTNPVFSFLNPLSPDIWMYILLA 570
sp | P42264 | GRIK3_RAT DLAVAPLTITHVREKAIDFSKPFMTLGVSIYRKPNGTNPVFSFLNPLSPDIWMYVLLA 572
sp | Q01812 | GRIK4_RAT DLAVAGLTTAEREKVIDFSKPFMTLGISILYRVHMGRRPYFSSLDPPFSPGVWLFMLLA 554
sp | Q63273 | GRIK5_RAT DLAVAAFTTTAEREKVIDFSKPFMTLGISILYRVHMGRKPGYFSLDPFSPAVWLFMLLA 553
***** : ** * * . ***** : ***** * . * * * : ** * : ** : **

sp | P22756 | GRIK1_RAT CLGVSCVLFVIARFTPYEWYNPHPCNPD-SDVVENNFTLLNSFWFGVGMALMQQGSSELMPK 644
sp | P42260 | GRIK2_RAT YLGVSCVLFVIARFSPYEWYNPHPCNPD-SDVVENNFTLLNSFWFGVGMALMQQGSSELMPK 629
sp | P42264 | GRIK3_RAT YLGVSCVLFVIARFSPYEWYDAHPCNPG-SEVVENNFTLLNSFWFGVGMALMQQGSSELMPK 631
sp | Q01812 | GRIK4_RAT YLAVSCVLFVLARLTPYEWYSPHPCAQGRCNLLVNQYSLGNSLWFPVGGFMQQGSTIAPR 614
sp | Q63273 | GRIK5_RAT YLAVSCVLFVLAARLSPYEWYNPHPCRLARPHILENQYTLGNSLWFPVGGFMQQGSEIMPR 613
* . ***** : ** : ***** . * * . : : * : ** * * * : * . : ***** : * :

sp | P22756 | GRIK1_RAT ALSTRIVGGIWWFFTLIIISSYTANLAAFLTVERMESPIDSAADDLAKQTKIEYGAVRDGS 704
sp | P42260 | GRIK2_RAT ALSTRIVGGIWWFFTLIIISSYTANLAAFLTVERMESPIDSAADDLAKQTKIEYGAVEDGA 689
sp | P42264 | GRIK3_RAT ALSTRIVGGIWWFFTLIIISSYTANLAAFLTVERMESPIDSAADDLAKQTKIEYGAVKDGA 691
sp | Q01812 | GRIK4_RAT ALSTRICVSGVWVAFTLIIISSYTANLAAFLTVQRMEVPIESVDDLADQTAIEYGTIHGGS 674
sp | Q63273 | GRIK5_RAT ALSTRICVSGVWVAFTLIIISSYTANLAAFLTVQRMEVPVESADDLADQTNIEYGTIHAGS 673
***** : * : ** ***** : ***** * : * . * * * . * * * * : . * :

sp | P22756 | GRIK1_RAT TMTFFKSKISTYKMWAFMSSRQQSALVKNSDEGIQRVLTDDYALLMESTSIEYVTQRN 764
sp | P42260 | GRIK2_RAT TMTFFKSKISTYDKMWAFMSSRRQSVLVKSNEEGIQRVLTSDYAFLMESTTIEFVTQRN 749
sp | P42264 | GRIK3_RAT TMTFFKSKISTFEKMWAFMSSKP-SALVKNEEGIQRVLTADYALLMESTTIEYITQRN 750
sp | Q01812 | GRIK4_RAT SMTFFQNSRYQTYQRMWNYMYSKQPSVFKSTEEGIARVLNSNYAFLESTMNEYRQRN 734
sp | Q63273 | GRIK5_RAT TMTFFQNSRYQTYQRMWNYMQSKQPSVFKSTEEGIARVLNSRYAFLESTMNEYHRLN 733
: ***** : : * : ** : ** * * : * : * . : * * * * * : * : * * : *

Asn751 (GluK2)

```

sp | P22756 | GRIK1_RAT      CNLTQIGGLIDSKGYGVGTPIGSPYRDKITIAILQLQEEGKLHMMKEKWWRGNGCPEEDS 824
sp | P42260 | GRIK2_RAT      CNLTQIGGLIDSKGYGVGTPMGSPYRDKITIAILQLQEEGKLHMMKEKWWRGNGCPEEES 809
sp | P42264 | GRIK3_RAT      CNLTQIGGLIDSKGYGIGTPMGSPYRDKITIAILQLQEEDKLHIMKEKWWRGSGCPEEEN 810
sp | Q01812 | GRIK4_RAT      CNLTQIGGLLDTKGYGIGMPVGSVFRDEFDLAILQLQENNRLEILKRKWWEGGKCPKEED 794
sp | Q63273 | GRIK5_RAT      CNLTQIGGLLDTKGYGIGMPLGSPFRDEITLAILQLQENNRLEILKRKWWEGGRCPKEED 793

*****:*****: *:* * :*: :*****: :*.:*.***.*. **:*.

sp | P22756 | GRIK1_RAT      KEASALGVENIGGIFIVLAAGLVLSVFVAIGEFLYKSRKNNDVEQKGKSSRLRFYFRNKV 884
sp | P42260 | GRIK2_RAT      KEASALGVQNIIGGIFIVLAAGLVLSVFVAVGEFLYKSKNAQLEKRSFCSAM----- 861
sp | P42264 | GRIK3_RAT      KEASALGIQKIGGIFIVLAAGLVLSVLVAVGEFIYKLRKTAEREQRSFCSTV----- 862
sp | Q01812 | GRIK4_RAT      HRAKGLGMENIGGIFVVLICGLLIVAFMAMLEFLWTLRHS-EASEVSVQCQEM----- 845
sp | Q63273 | GRIK5_RAT      HRAKGLGMENIGGIFVVLICGLIIAVFVAVMEFIWSTRRSAESEVSVQCQEM----- 845

:.*.***:*****:* .***:*****: **: :. :. :. :. :

sp | P22756 | GRIK1_RAT      RFHGSKKESLGVKCLSFNAIMHEELGISLKNQKLLKKSRT-----KGKSSFT---S 933
sp | P42260 | GRIK2_RAT      -----VEELRMSLKCQRRLKHKPQA-----PVIVKTEEVIN 892
sp | P42264 | GRIK3_RAT      -----ADEIRFSLTCQRRLKHKPQP-----PMMVKTDAVIN 893
sp | Q01812 | GRIK4_RAT      -----MTELRSIILCQDNIHPRRRRSGGLPPQPPVLEERRPRGT 884
sp | Q63273 | GRIK5_RAT      -----LQELRHAVSCRKTSRSRRRRRPGGPRALL--SLRAVRE 882

* : : : : :

sp | P22756 | GRIK1_RAT      ILTCHQRRT---QRKE-----TV---A----- 949
sp | P42260 | GRIK2_RAT      MHTFNDRRL---PGKE-----TM---A----- 908
sp | P42264 | GRIK3_RAT      MHTFNDRRL---PGKD-----SM---SCSTSLAPVFP----- 919
sp | Q01812 | GRIK4_RAT      ATLSNGKLCGAGEPD-----QLAQL---AQEAALVARGCTHIRVCECRRFQGLR 932
sp | Q63273 | GRIK5_RAT      MRLSNGKLYSAGAGGDAGAHGGPQRLLDDPGPPGGPRPQAPTPCTHVRVCQECRRIQALR 942

: : :

sp | P22756 | GRIK1_RAT      ----- 949
sp | P42260 | GRIK2_RAT      ----- 908
sp | P42264 | GRIK3_RAT      ----- 919
sp | Q01812 | GRIK4_RAT      ARSPARSEESLEWDKTTNSSEPE----- 956
sp | Q63273 | GRIK5_RAT      ASGAGAPPRGLGTPAEATSPPRPRPGPTGPRELTEHE 979

```

The N-GlycoSite program (<https://www.hiv.lanl.gov/content/sequence/GLYCOSITE/glycosite.html>) predicts 9 glycosylation sites in GluK1-3, 8 sites in GluK4 and 10 sites in GluK5. We observed carbohydrate density for all 9 sites in GluK2 and built anywhere from 0 to 8 (depending on structure) sugar moieties per site. The residues N378, N546 and N751 involved in interaction with ConA are highly conserved in GluK1-3 but not in GluK4-5. Given that GluK4-5 cannot form homotetramers and can only form heterotetramers with GluK1-3, any KAR must include these three glycosylation sites. Therefore, there is a possibility that ConA can bind to any KAR. On the other hand, regulation of KARs by ConA may still be specific to certain KAR subunits or their combination and depend on the local structural environment. While this question is interesting, to properly study the GluK2 specificity toward ConA would require electrophysiological experiments with different KAR subunits and their assemblies, which is outside the scope of the present work. As the N378, N546 and N751 glycosylation sites individually do not play an apparent role in expression (likely because the other 6 sites can “back them up” in this function), what is their role in the absence of ConA also remains an open question that would require additional study.

2. ConA and BPAM together markedly slow down the KAR desensitization, however, the individual roles of ConA and BPAM and their cooperativity in KAR activation remain ambiguous from the structural analysis in the manuscript. A more detailed explanation and discussion would be helpful.

Reviewer #1 is right that the reason for cooperativity of ConA and BPAM in KAR activation that is apparent from functional recordings (Fig. 1b) is not obvious from our structures. When we attempted to open the channel by Glu (~30-s application using regular Vitrobot) in the presence of BPAM alone (no ConA added), the only conformation we got was GluK2_{Glu-4fold}, indicating that BPAM alone cannot hold the D1-D1 interfaces intact on the time scale of > 1 s (Fig. 1b, second trace) and Glu binding causes dissociation of BPAM and breakdown of LBD dimers. We think that ~100° rotation of B/D LBDs that accompany the breakdown of LBD dimers is sterically hindered by ConA molecules bound to the D2 lobes of LBDs. The corresponding hypothesis has now been discussed in the text (lines 243-247).

3. The manuscript described two structures of GluK2 without ConA binding in symmetric and asymmetric forms. Despite the addition of BPAM, no BPAM was observed, obviously caused by the abruption of the BPAM site in LBD. However, the previous findings from the same lab showed the BPAM molecule in the GluK2 structure. Could the authors elaborate on the distinctions between these structures and discuss the potential reasons of missing a class with BPAM only?

Previously we solved structures of GluK2 in the presence of BPAM but in the absence of Glu (Gangwar et al., 2023). In these conditions, the LBD clamshells remain open and BPAM can easily stabilize the D1-D1 interface. When we subject GluK2 to cryo-EM in the presence of BPAM and Glu, we only get GluK2_{Glu-4fold} structure (see the previous comment), indicating that Glu-induced closure of LBD clamshells causes rupture of the D1-D1 interfaces, thus destroying the BPAM binding sites. Accordingly, the presence of BPAM (with no ConA added) does not prevent the D1-D1 interface rupture, likely because the “BPAM glue” is not strong enough against Glu action. In this study, we

present structures that have been solved in the presence of Glu (< 3-s long application). Accordingly, those particles that have ConA bound reveal binding of BPAM because ConA helps BPAM to maintain the D1-D1 interfaces intact. For those particles that have no ConA bound, the LBD dimers undergo the D1-D1 interface rupture under action of Glu. On the time scale of our experiment, the action of Glu is fast (Fig. 1b, second trace) and all particles on the grid already do not have BPAM bound. Nevertheless, some particles still have the D2-D2 interface intact, contributing to the GluK2_{Glu-asym} conformation, while others have LBD dimers completely dissociated, resulting in GluK2_{Glu-4fold} conformation. The corresponding explanation has been added to the text (lines 206-220, 243-247).

4. The transmembrane domains and lipids in the two GluK2 structures without ConA are either missing or not clearly defined. However, the previously published GluK2 structures have well-defined TMD and lipids. Could the authors provide an explanation for the disordered transmembrane domain in this state?

Previous structures of GluK2 alone without ConA have clearly defined TMD and lipids only in the absence of agonist and presence of BPAM (PDB ID: 8FWS, 8FWU, 8FWW). KAR structures in the presence of agonist do generally appear to have more flexible TMD. The only occasion when TMD appeared to be stabilized enough to see one annular lipid is in the presence of the auxiliary subunit Neto (PDB ID: 7F5B; only one lipid per GluK2 tetramer!). For all other structures solved in the presence of agonist (e.g., PDB ID: 5KUF, 7KS3), lipids are not resolved. One possible explanation for such TMD behavior is that most published KAR structures solved in the presence of agonists are structural artifacts, with LBD dimers dissociating into monomers due to specific forces applied to receptors associated with air-liquid interface of the cryo-EM grid (lines 282-290).

5. The ATD-LBD linker exhibits flexibility to accommodate regulatory molecules like ConA. How the presence or absence of ConA binding affects the linker. Specifically, how does ConA binding alter the structure of dynamics of the ATD-LBD linker, and what implication does this have for channel modulation?

In the ConA bound states, the ATD-LBD linkers become stretched out to accommodate the increased distance between the ATD and LBD layers. Despite being stretched out, these linkers have a capacity to stretch out even more because in the linearized form they will be longer. Correspondingly, with restraints on their conformations present only at the points of attachment to ATD and LBD, their middle portions remain flexible and poorly visible in cryo-EM maps (Fig. 1c, Extended Data Fig. 1). In the absence of ConA, the ATDs and LBDs in KAR subunits A and C form interfaces that determine the most stable conformations of ATD-LBD linkers, which wrap around these interfaces and appear most visible in cryo-EM maps. At the same time, there are no restraints on the ATD-LBD linkers in subunits B and D, except their attachment points to ATD and LBD, what makes them flexible in the middle and poorly visible in cryo-EM maps. Overall, the ATD-LBD linkers appear to adapt their conformations and the major players that determine channel modulation remain to be the agonists and positive allosteric modulators (lines 239-251).

6. Two types of ConA-LBD interaction are observed, type I and type II (Fig. 2f and 2g). However, it is unclear how these two types are defined in the structure as the figure 2a only shows interaction in Figure 2f. Are they from two different classes? How the interacting residues such as L99, Y100 or R228 in ConA affect the modulation effect on GluK2?

Fig. 2a shows the structure for the most populated $\text{GluK2}_{\text{Glu-2xConA-BPAM}}$ class (see Extended Data Fig. 1) with ATD-ConA interface closeup view in Fig. 2e and type I ConA-LBD interface closeup view in Fig. 2f. To clarify which types of ConA-LBD interfaces are present in the five different classes in complex with ConA (two with one ConA and three with two ConA), we added labels indicating the types of ConA-LBD interfaces for each class (1: type I; 2: type II; 3: type I + type I; 4: type I + type II; 5: type II + type II) in Extended Data Fig. 1. For all three types of interfaces, the residues on the ConA side appear to be the same, indicating that they assemble the major site of ConA interaction with carbohydrates. The precise role of individual ConA residues (L99, Y100 and R228) in modulation of GluK2 function would require point mutagenesis, which would be difficult to do, since we do not produce ConA ourselves and instead use a commercially available ConA. Such study would be an interesting topic for a separate investigation.

7. What are the differences between 1x-ConA and 2x-ConA structure? It is not evident from the current description if both configurations lead to open states or if they differ in their pore profiles (Fig 4). Additionally, in the $\text{GluK2}_{\text{Glu-1xConA-BPAM}}$ structure, it would be beneficial to understand how ConA's presence influences BPAM binding. Does the loss of one ConA molecule alter the LBD configuration compared to the $\text{GluK2}_{\text{Glu-2xConA-BPAM}}$ structure? Is one ConA sufficient to open the channel?

For the LBD-TMD region (including the LBD configuration), there is no difference between $\text{GluK2}_{\text{Glu-1xConA-BPAM}}$ and $\text{GluK2}_{\text{Glu-2xConA-BPAM}}$ structures. This was the reason why we combined particles representing both structures to get the best resolution of the LBD-TMD region (Extended Data Fig. 1). This obviously means that the presence of just one bound ConA molecule is sufficient (1) for full occupancy of the BPAM sites and (2) to open the channel. We have emphasized these observations in the text (lines 161-166).

Minor:

1. It is unclear about the functional state of the asymmetrical $\text{GluK2}_{\text{Glu}}$ structure, does it undergo a channel desensitization from open state induced by Glu, ConA and BPAM?

We think that the kinetic scheme for KARs is in general similar to AMPA receptors (Twomey and Sobolevsky, Biochemistry 2018). Accordingly, after Glu binds, KAR enters a pre-active state first. From the pre-active state, it can either open (very briefly in the absence of PAMs and for somewhat longer time in the presence of PAMs BPAM and ConA) or desensitize (in this case PAMs are not bound). The second route (desensitization) for KARs appears to happen in extended fashion compared to AMPA receptors: (1) the D1-D1 interfaces undergo rupture but the D2-D2 interfaces stay intact (this is how desensitization in AMPA receptors occurs) but then (2) one LBD dimer

become dissociated (GluK2_{Glu-*asym*}) and (3) the second LBD dimer becomes dissociated as well (GluK2_{Glu-*4fold*}). The corresponding explanations have been added to the Discussion (lines 252-253, 267-276)

2. Keep the nomenclature of BPAM in consistency throughout the text. E.g. Line 103: “Given the synergistic effect of PAMs on the current amplitude”.

We use abbreviation PAM for Positive Allosteric Modulator (line 73) and BPAM for the specific positive allosteric modulator BPAM344 (line 78). We have carefully checked our abbreviations throughout the text to make sure that they are consistent.

3. Extended Data Fig. 3. Densities of glycosylation sites and carbohydrates linked to ConA binding should be shown.

We have added densities for the glycosylation sites and carbohydrates linked to ConA as new panels c-e in Extended Data Fig. 3.

Referee #2 (Remarks to the Author):

This is a neat paper with great figures that expands our view of glutamate receptor activation. Using an elegant time-resolved approach calibrated to functional measurements, the authors have managed to catch the GluK2 kainate receptor in a variety of activation states, including, against the odds, an open state. Technical challenges are, as usual with this expert group, met and overcome.

We thank Reviewer #2 for the generous comments on our work.

Major points

1. What is new here? We get a new open arrangement, the first in kainate receptors, and it's a bit different from AMPA receptors. The authors conclude that this open kainate receptor channel must conduct ions and water because AMPA receptors that are open do so in MD simulations. I thought it's rather a shame not to show this simple result computationally (if indeed that is what happens). Not all activated AMPA channels conduct in MD simulations, even those that look as if they really should, so this is not really an open and shut case. Collapsing the pore dimensions to a one dimensional profile is often done but doesn't really answer this point. The current presentation of this data doesn't do a good job of bringing out the novelty.

We have now run MD simulations for the closed GluK2_{BPAM} (PDB ID: 8FWS) and open GluK2_{Glu-ConA-BPAM} structures and showed that the first one does not conduct water and ions, while the second one conducts both (Extended Data Fig. 8).

2. The authors do update our understanding of the lectin Concanavalin A considerably, but do not

reference previous work that would indicate the extent of the advance. In somewhat implausible but influential work, it was shown that one glycan, any glycan -even an ectopic site -, was all that was needed for the activity of ConA (Everts I, Petroski R, Kizelsztejn P, Teichberg VI, Heinemann SF, Hollmann M (1999) Lectin-induced inhibition of desensitization of the kainate receptor GluR6 depends on the activation state and can be mediated by a single native or ectopic N-linked carbohydrate side chain. J Neurosci 19(3):916–927.)^[SEP] This study was odd because it suggested instead of a clear structural mechanism (as demonstrated here) rather something non-specific breaking the receptor desensitization, perhaps because ConA is a “dirty” ligand. It would be good to cite this work (and perhaps later work from Derek Bowie) that would indicate the confusion and difficulty in this area. The work in hand does a lot to clear up this confusing picture, and for the first time explains how ConA works.

We thank Reviewer #2 for appreciating our work. We have now cited the work of Hollmann’s, Bowie’s and Jayaraman’s groups on ConA regulation of KARs (lines 80-84).

3. There are plenty of inactive state structures already known for other glutamate receptors, so the menagerie of extra structures provided here is merely welcome, rather than revolutionary.^[SEP]

Indeed, structures similar to one of the two inactive state structures that we present (GluK2_{Glu-4fold}) have already been published (e.g., PDB ID: 5KUF). We present GluK2_{Glu-4fold} very briefly to give a complete description of the ensemble of conformations observed in our structural experiment. The other inactive state (GluK2_{Glu-asym}), however, is unique and similar structures have not been published before. We do consider this structure revolutionary, because it shows for the first time an intermediate desensitized state, which suggests that at least initial stages of the desensitization process in KARs occur similarly to AMPA receptors. The reason why we think this conclusion is revolutionary is because previously KAR desensitization was declared to happen in a completely different from AMPA receptors way (Meyerson et al nature 2016, Khanra et al Elife 2021, He et al Nature 2021).

4. The authors write that “Our structures reveal the molecular basis of KAR gating that determines the unique role of these receptors in neurobiology and disease” which is hyperbole built on exaggeration. I didn't find anything in this paper to help in this respect. We could imagine from the new structures how to target kainate receptors with novel therapeutics, which the authors mention. The provided list of neurological disorders that kainate receptors have been linked to is long, and the authors reference some reviews on the topic. I think there are as many reviews as original research works in this area. And most of the reviews feature a lot of synaptic physiology and then some “maybe” sections at the end about pathological states. Most of the genetics studies have not been followed up. I know that it would be nice to think that these structures explain kainate receptors in the brain, but they don't add much to an already very scarce pool of information. A particular concern in this regard is that almost no receptors in the brain are GluK2 homomers, so the connection to physiology or pathology is even more tenuous. Heteromer structures of kainate receptors were already published.

We agree with Reviewer #2 that the majority of KARs in the brain are heterotetramers. To our defense, not a single heteromeric structure of AMPAR or KAR showed any novel structural

mechanism that would not be discovered for the homotetrameric AMPARs or KARs before. Besides, running structural experiments with heteromeric receptors is generally more difficult because of lower expression and increased sample heterogeneity. Nevertheless, we have changed the mentioned sentence so that it now reads “Our structures reveal the molecular basis of KAR gating that may guide the development of drugs for treatment of neurological disorders”.

5. The authors cite two open access designs for plungers, one that is light coupled. Perhaps the authors could give a bit more detail? The methods section does not read naturally and is, to my eye, inadequate.

“The operational principles of the in-house vitrification system align with a time-resolved vitrification device(45,46) previously outlined with minor modifications.”

One could also argue that the time-resolved aspect is somewhat overplayed, coming in at 3 seconds. But, the functional measurements show that this is an appropriate timescale. In any case, the technological aspect here is the most important, and should be described more completely.

We expanded the section of Methods that describes our custom plunger (lines 554-565). We have also added an illustration of this plunger as new Extended Data Fig. 10.

Minor points

* The authors do not mention the RNA editing status of the construct.

We have now mentioned that we have used rat GluK2 with V at position 567, C at position 571 and Q at position 621 (the Q/R site) (lines 514-516).

* 3 references (21-23) are cited in a block of 11 at the end of the following sentence

“Both principal and auxiliary subunits determine KAR functional properties, including ion permeation, channel block, ligand specificity and kinetics, and play important roles in shaping KAR-mediated synaptic responses”

These three references, all from Derek Bowie’s lab, are about ion modulation and don’t seem to bear on the properties discussed. They were largely superseded by structures of the ion binding sites. But maybe the authors find particular value in these works that I missed.

The three references mentioned by Reviewer #2 have now been removed from the text.

Reviewer Reports on the First Revision:

Referees' comments:

Referee #1 (Remarks to the Author):

The authors have addressed all my concerns and questions in the revised version. I recommend the publication in Nature. An optional comment: the authors may consider to add the sequence alignment of GluK2 with other kainate receptor subtypes in the supplementary figure, highlighting the conserved residues for ConA binding and glycosylation sites.

Referee #2 (Remarks to the Author):

The authors have made a constructive response and I think the paper is better for the improved context. All the points from my previous review (and that of the other reviewer) have, in my opinion, been well answered.

Major points

The molecular dynamics simulations are a good addition. The technical setup is mostly clear and seems appropriate. But some details are missing to understand the conditions of the simulations. Was an electric field applied to drive the ions through the pore? If the simulation was done at 0 mV transmembrane potential, then it is hard to imagine why the ions pass through in the direction of the intracellular space. In the video, ions traverse the entire channel and there is a net transfer of one ion from the outside to the inside. No single ion undergoes an entire transit of the pore.

In contrast to the video, there is no analysis in the figure to show that ions pass the gate region, which is perhaps underplaying the results. ^[1111]_[SEPISEP]In the methods it says how the trajectories were analysed, but we don't see any analysis of trajectories as such. We only see the "cumulative distribution of ions", which is plotted in a crude way. I understand that this is a first look, but this presentation is not so helpful at all, and I would say it's at a lower level of quality than the rest of the paper. It would be better to plot contours of ion abundance.

—

I am sorry, but when re-reading, I noticed two points in the same paragraph which were presumably present in the original submission, but that I overlooked before. I think they are eventually important and should be fixed with changes to the text.

Paragraph beginning line 252 : ^[1111]_[SEPISEP]"KAR activation appears similar to that described in AMPARs and likely occurs through a pre-active state, in which Glu has already bound but the channel is not yet open^{35,38,39,51}. "

There are a lot of pre-open, ligand bound states published for all iGluRs, including AMPA and NMDA.

It would be worth a mention, to emphasise the similarities. I am not sure if the right references have been put here. There were a bunch of pre-open states for AMPAR published in 2014 from crystal structures (e.g. Dürr et al 2014, Yelshanskaya et al 2014). Are these not the relevant ones? Only the Chen_2017 cell paper (your ref #39) has a kainate pre-open structure? I do understand, this point is discussed in reference 51. I'm talking about the original papers. Perhaps I am missing the point that you want to make.

“One possible explanation is that among open states with different ion conductance (O1-O4, with increasing channel opening), GluK2Glu-ConA-BPAM represents the maximal conductance O4 state, while AMPAR structures represent intermediate conductance O1 and O2 states⁴⁶.”

Leaving aside the structural interpretation of AMPA sublevels, which in my view is still inconclusive (the authors may well disagree!), I do not know of evidence for KARs having the same O1-O4 staircase as AMPARs, with sublevels corresponding to increasing numbers of agonist bound. NMDARs do not have this behaviour. It will perhaps be possible with a combination of BPAM and ConA to measure this subconductance gating in KARs in the future, but until then, it is equally plausible that KAR like NMDAR have only one main conductance level (and a sublevel not related to subunit-wise activation). Therefore, a direct relation to the status of the 5WEO structure, or this group's other work on AMPA sublevels, remains unclear.

I think that the sentence cited above is fine, but you ought to state afterwards, that the O1-O4 staircase is not established for KARs, which leaves this point open.

Author Rebuttals to First Revision:

We thank Reviewers for their additional suggestions. Addressing these suggestions has further improved our manuscript, with the details outlined in our responses below.

Referee #1 (Remarks to the Author):

The authors have addressed all my concerns and questions in the revised version. I recommend the publication in Nature. An optional comment: the authors may consider to add the sequence alignment of GluK2 with other kainate receptor subtypes in the supplementary figure, highlighting the conserved residues for ConA binding and glycosylation sites.

Per Reviewer #1 request, we have added an alignment of KAR subunits as a new Supplementary Figure 1 (lines 144-145, 148-151, 933-936).

Referee #2 (Remarks to the Author):

The authors have made a constructive response and I think the paper is better for the improved context. All the points from my previous review (and that of the other reviewer) have, in my opinion, been well answered.

Major points

The molecular dynamics simulations are a good addition. The technical setup is mostly clear and seems appropriate. But some details are missing to understand the conditions of the simulations. Was an electric field applied to drive the ions through the pore? If the simulation was done at 0 mV transmembrane potential, then it is hard to imagine why the ions pass through in the direction of the intracellular space. In the video, ions traverse the entire channel and there is a net transfer of one ion from the outside to the inside. No single ion undergoes an entire transit of the pore.

In contrast to the video, there is no analysis in the figure to show that ions pass the gate region, which is perhaps underplaying the results. In the methods it says how the trajectories were analysed, but we don't see any analysis of trajectories as such.

We agree with Reviewer #2 that our description of the MD simulations and the presentation of MD results were somewhat lacking details and clarity. In the revised manuscript, we extended the presentation of the MD analysis and expanded Extended Data Fig. 8 to include panel **b** that shows the RMSD plot for the simulated structures and demonstrates the stability of simulations, and panel **g** that illustrates dynamics of several ions inside the channel. Panel **g** also demonstrates that despite the lack of net current in the absence of applied voltage, the channel gate is open and permeable. Finally, to clearly illustrate ion permeation, we updated the Supplementary Video 1 by making it longer (for the duration of simulation illustrated in Extended Data Fig. 8g).

We have amended and clarified the text of the manuscript to better describe the simulations and their results (lines 190-195, 258, 719-720, 891-908, 938-947). In brief, the simulations were

performed in the equilibrium manner with no applied voltage as correctly guessed by Reviewer #2. In these conditions, the net ion current was neither expected nor observed. The goal of our simulations was to demonstrate that the channel gate is open in GluK2_{Glu-ConA-BPAM} and closed in GluK2_{BPAM}. To show this, it is sufficient to illustrate hydration of the gate region (Extended Data Fig. 8c,d) and ability of ions to cross the gate in either direction (Extended Data Fig. 8e,f,g). During our relatively short simulations of GluK2_{Glu-ConA-BPAM}, we observed several events when ions crossed the gate region between the central cavity and extracellular vestibule in both directions, confirming that the gate in GluK2_{Glu-ConA-BPAM} is open. In contrast, no ion permeation or water hydration was observed in GluK2_{BPAM}.

We only see the “cumulative distribution of ions”, which is plotted in a crude way. I understand that this is a first look, but this presentation is not so helpful at all, and I would say it’s at a lower level of quality than the rest of the paper. It would be better to plot contours of ion abundance.

As requested by the Reviewer #2, we have changed panels d and f in Extended Data Fig. 8 to show the contours of ion abundance instead of the ion cumulative distribution.

—

I am sorry, but when re-reading, I noticed two points in the same paragraph which were presumably present in the original submission, but that I overlooked before. I think they are eventually important and should be fixed with changes to the text.

Paragraph beginning line 252 : “KAR activation appears similar to that described in AMPARs and likely occurs through a pre-active state, in which Glu has already bound but the channel is not yet open^{35,38,39,51}. “

There are a lot of pre-open, ligand bound states published for all iGluRs, including AMPA and NMDA. It would be worth a mention, to emphasise the similarities. I am not sure if the right references have been put here. There were a bunch of pre-open states for AMPAR published in 2014 from crystal structures (e.g. Dürr et al 2014, Yelshanskaya et al 2014). Are these not the relevant ones? Only the Chen_2017 cell paper (your ref #39) has a kainate pre-open structure? I do understand, this point is discussed in reference 51. I’m talking about the original papers. Perhaps I am missing the point that you want to make.

Reviewer #2 is right – Dürr et al 2014, Yelshanskaya et al 2014 describe the putative pre-open conformations of AMPARs. The reference 51 review was cited to reduce the total number of citations. We have now included citations of the original papers.

“One possible explanation is that among open states with different ion conductance (O1-O4, with increasing channel opening), GluK2_{Glu-ConA-BPAM} represents the maximal conductance O4 state, while AMPAR structures represent intermediate conductance O1 and O2 states⁴⁶.”

Leaving aside the structural interpretation of AMPA sublevels, which in my view is still inconclusive (the authors may well disagree!), I do not know of evidence for KARs having the same O1-O4

staircase as AMPARs, with sublevels corresponding to increasing numbers of agonist bound. NMDARs do not have this behaviour. It will perhaps be possible with a combination of BPAM and ConA to measure this subconductance gating in KARs in the future, but until then, it is equally plausible that KAR like NMDAR have only one main conductance level (and a sublevel not related to subunit-wise activation). Therefore, a direct relation to the status of the 5WEO structure, or this group's other work on AMPA sublevels, remains unclear.

I think that the sentence cited above is fine, but you ought to state afterwards, that the O1-O4 staircase is not established for KARs, which leaves this point open.

Similar to AMPARs, KARs show multiple conductance levels, which was documented in several publications, including the following examples¹⁻⁴:

- 1 Swanson, G. T., Feldmeyer, D., Kaneda, M. & Cull-Candy, S. G. Effect of RNA editing and subunit co-assembly on single-channel properties of recombinant kainate receptors. *J. Physiology* **492.1**, 129-142 (1996).
- 2 Zhang, W. *et al.* A transmembrane accessory subunit that modulates kainate-type glutamate receptors. *Neuron* **61**, 385-396 (2009).
- 3 Zhang, W., Devi, S. P., Tomita, S. & Howe, J. R. Auxiliary proteins promote modal gating of AMPA- and kainate-type glutamate receptors. *Eur J Neurosci* **39**, 1138-1147, doi:10.1111/ejn.12519 (2014).
- 4 Howe, J. R. Modulation of non-NMDA receptor gating by auxiliary subunits. *J Physiol* **593**, 61-72, doi:10.1113/jphysiol.2014.273904 (2015).